# A retinoic acid-dependent stroma-leukemia crosstalk promotes chronic lymphocytic leukemia progression

Diego Farinello[1], Monika Wozińska[1], Elisa Lenti[1], Luca Genovese[1], Silvia Bianchessi[1], Edoardo Migliori[1], Nicolò Sacchetti[1], Alessia di Lillo[1], Maria Teresa Sabrina Bertilaccio[1,10], Claudia de Lalla[2], Roberta Valsecchi[1], Sabrina Bascones Gleave[3], David Lligé[3], Cristina Scielzo[1], Laura Mauri[4], Maria Grazia Ciampa[4], Lydia Scarfò[1], Rosa Bernardi[1], Dejan Lazarevic ⓘ [5], Blanca Gonzalez-Farre[6], Lucia Bongiovanni[7], Elias Campo[6], Andrea Cerutti[3], Maurilio Ponzoni[7,8], Linda Pattini ⓘ [9], Federico Caligaris-Cappio[1,11], Paolo Ghia ⓘ [1,8] & Andrea Brendolan[1]

In chronic lymphocytic leukemia (CLL), the non-hematopoietic stromal microenvironment plays a critical role in promoting tumor cell recruitment, activation, survival, and expansion. However, the nature of the stromal cells and molecular pathways involved remain largely unknown. Here, we demonstrate that leukemic B lymphocytes induce the activation of retinoid acid synthesis and signaling in the microenvironment. Inhibition of RA-signaling in stromal cells causes deregulation of genes associated with adhesion, tissue organization and chemokine secretion including the B-cell chemokine CXCL13. Notably, reducing retinoic acid precursors from the diet or inhibiting RA-signaling through retinoid-antagonist therapy prolong survival by preventing dissemination of leukemia cells into lymphoid tissues. Furthermore, mouse and human leukemia cells could be distinguished from normal B-cells by their increased expression of *Rarγ2* and *RXRα,* respectively. These findings establish a role for retinoids in murine CLL pathogenesis, and provide new therapeutic strategies to target the microenvironment and to control disease progression.

[1] Division of Experimental Oncology, IRCCS, San Raffaele Scientific Institute, Milan, 20132, Italy. [2] Division of Immunology and Infection Disease, IRCCS, San Raffaele Scientific Institute, Milan, 20132, Italy. [3] Program for Inflammatory and Cardiovascular Disorders, Institut Hospital del Mar d'Investigacions Mèdiques (IMIM), Barcelona, 08003, Spain. [4] Department of Medical Biotechnology and Translational Medicine, University of Milan, Milano, 20090, Italy. [5] Center of Translational Genomics and Bioinformatics, IRCCS, San Raffaele Scientific Institute, 20132 Milan, Italy. [6] Institut d'Investigacions Biomèdiques August Pi i Sunyer (IDIBAPS), Hospital Clinic and University of Barcelona, Calle Roselló 149-153, Barcelona, 08036, Spain. [7] Pathology Unit, IRCCS San Raffaele Scientific Institute, Milan, 20132, Italy. [8] Vita-Salute San Raffaele University School of Medicine, Milan, 20132, Italy. [9] Department of Electronics, Information and Bioengineering, Politecnico di Milano, Milan, 20133, Italy. [10]Present address: Department of Experimental Therapeutics, The University of Texas MD Anderson Cancer Center, Houston, 77054 TX, USA. [11]Present address: Associazione Italiana per la Ricerca sul Cancro, Via San Vito 7, Milano, 20123, Italy. These authors contributed equally: Diego Farinello, Monika Wozińska, Elisa Lenti. Correspondence and requests for materials should be addressed to A.B. (email: brendolan.andrea@hsr.it)

Chronic lymphocytic leukemia (CLL), the most frequent adult leukemia in Western countries, is characterized by the expansion of mature CD5[+] B cells in protective microenvironmental niches of secondary lymphoid organs (SLOs) and bone marrow (BM). In these tissues, the interactions between leukemia and cells of the microenvironment promote tumor cell survival, chemoresistance, and disease progression[1–3]. The non-hematopoietic compartment of SLOs comprises different stromal cell subsets including follicular stromal cells, whose role in CLL pathogenesis is still largely unknown[4–7]. Understanding how the stromal compartment evolves and which molecular pathways are involved in supporting tumor cell survival and expansion is crucial to elucidate the contribution of stromal cells in CLL pathogenesis and to design novel therapeutic strategies aiming to target stromal microenvironmental interactions. Stromal cells play a crucial role in organizing lymphoid compartments and in regulating lymphoid homeostasis through the secretion of chemokines and the deposition of the extracellular matrix (ECM), a tri-dimensional scaffold that supports adhesion and locomotion of normal and malignant lymphocytes and acts as a reservoir of signaling molecules and growth factors[8–11]. Aberrant stromal remodeling has been also differentially associated with lymphoid malignancies, including CLL; although the molecular mechanisms underlying it remain elusive.

Retinoic acid (RA), the active metabolite of Vitamin A, is an essential molecule required for vertebrate development and tissue homeostasis[12–15]. RA binds to nuclear receptors and regulates numerous biological processes including cellular differentiation, adhesion, migration, and tissue remodeling[16–19]. In cancer, retinoids and their synthetic analogs are used in the pre-clinical and clinical settings for the treatment of hematologic malignancies and other types of cancer with the rational to induce terminal differentiation and/or apoptosis[20,21]. On the contrary, emerging data indicate that genetic ablation of RA-nuclear receptors or administration of retinoid-antagonist therapy has also been effective in pre-clinical models of breast cancer, allograft rejection, and myelofibrosis, although these approaches have not yet been reported in clinical setting or for the treatment of lymphoid malignancies. Contrary to the pro-differentiation effect of retinoid-analogs, the inhibition of RA-signaling was shown to affect multiple pathways ranging from reduced chemokine secretion, lymphocyte migration, and stromal remodeling[22–24].

Herein, we set out to characterize the evolution of the stromal microenvironment during CLL progression and identify the molecular pathways involved. We show that leukemia induces RA synthesis and signaling in the stromal microenvironment, and that inhibition of RA-signaling in stromal cells affects genes associated with adhesion, tissue organization, and chemokine secretion. We further demonstrate that blocking RA-signaling controls disease progression and prolongs survival, thus opening to novel potential therapeutic strategies to treat CLL by targeting stroma–leukemia interactions through inhibition of retinoid signaling.

## Results

**Leukemia induces tissue remodeling and retinoid metabolism.** Recent work in mice demonstrated that few hours after injection into wild-type recipients, Eµ-TCL1 CLL cells migrate to follicles in a CXCR5-dependent manner and engage a cross-talk with follicular stromal cells via LTβR, resulting in CXCL13 secretion by stromal cells, leukemia activation, and proliferation[25]. To investigate the molecular pathways activated upon stroma-leukemia cross-talk, including those implicated in chemokine secretion, we performed a microarray analysis using mRNA purified from a murine spleen stromal cell line (mSSC) cultured

for 48 h with either murine Eµ-TCL1 CLL cells or control splenic B cells (Fig. 1a). Up-regulated transcripts in stromal cells cultured with Eµ-TCL1 CLL cells revealed significant enrichment for interferon regulatory factor (IRF) targets, genes related to extracellular region, exosomes, and inflammatory responses (Fig. 1a and Supplementary Fig. 1). Up-regulated IRF targets contain the bone marrow stromal cell antigen 2 (Bst2) gene, a membrane protein overexpressed in cancer (Supplementary Fig. 1)[26]. We also found the over-expression of actin alpha 2 smooth muscle (Acta2/αSma), a mesenchymal marker characteristic of cancer associated fibroblasts (Supplementary Fig. 1)[27]. Among deregulated genes annotated for extracellular region, we also discovered the up-regulation of Aldh1a1, Cyp1b1, and Aldh3b1, all genes encoding for RA-synthetizing enzymes (Supplementary Fig. 1). Differentially expressed genes were overall found significantly enriched for extracellular matrix (ECM) components annotated in the Matrisome-DB atlas, which represents an effort toward the characterization of global composition of the ECM[28]. They are categorized in core matrisome genes, comprising ECM glycoproteins, collagens and proteoglycans, and ECM-associated proteins including ECM-affiliated proteins, ECM regulators and secreted factors (Fig. 1a). In addition, we found down-regulation of gene-signatures related to cell cycle and cell division, indicating that leukemic cells do not promote stromal cell proliferation (Fig. 1a). To test if human CLL cells induce similar changes in stromal cells, we cultured human leukemic cells, negatively purified from the peripheral blood of eight CLL patients with stable disease, with the mSSC line for 24 h. qPRC analysis revealed a differential induction of genes belonging to retinoid synthesis (Aldh1a1, Cyp1b1, and Rdh10), fibroblast activation (αSma), and ECM (Prelp, Lamininβ2, Nidogen2) (Supplementary Fig. 2a). Previous work by Paggetti et al., showed that human CLL-derived exosomes induce the transition of stromal cells into αSMA[+] cancer-associated fibroblasts[29], a phenotype that is, at least in part, induced in stromal cells by Eµ-TCL1 CLL cells (Supplementary Fig. 1). The re-analysis of the dataset published by Paggetti revealed induction in stromal cells of genes belonging to inflammatory process, interferons, and cell cycle, all signatures that we also found deregulated in stromal cells upon mouse leukemic cell culture. Notably, of the commonly expressed genes in the two datasets, a large fraction was similarly up-regulated in stromal cells (Supplementary Fig. 2b).

To confirm that Eµ-TCL1 CLL cells can indeed modulate the RA-signaling pathway in the microenvironment, we exploited an in vitro system, in which F9 cells, expressing LacZ under the RA responsive elements (RARE) are cultured with leukemic or control B cells (Fig. 1b). We found that βgal activity, indicative of the RA-signaling activation, was significantly higher in the presence of leukemic, as compared to control B-cells (Fig. 1b). To confirm that this effect was RA-dependent, we treated the cultures with the RA-signaling inhibitor BMS493[30], and found that this significantly abrogates βgal activity, and thus RA-signaling (Fig. 1b, right panel).

We then assessed whether modulation of RA-activity in responder cells could be the consequence of a paracrine effect of retinoids secreted by Eµ-TCL1 CLL cells. To this end, we performed Aldefluor staining to determine the Aldehyde dehydrogenase (ALDH) activity that is required to produce retinoic acid. We found that a fraction of Eµ-TCL1 CLL cells possesses ALDH activity, a phenotype that was specifically abrogated using the ALDH inhibitor DEAB (Supplementary Fig. 3).

**Inhibiting RA-signaling prevents CLL-induced gene expression.** We then assessed the effect of RA-signaling inhibition in

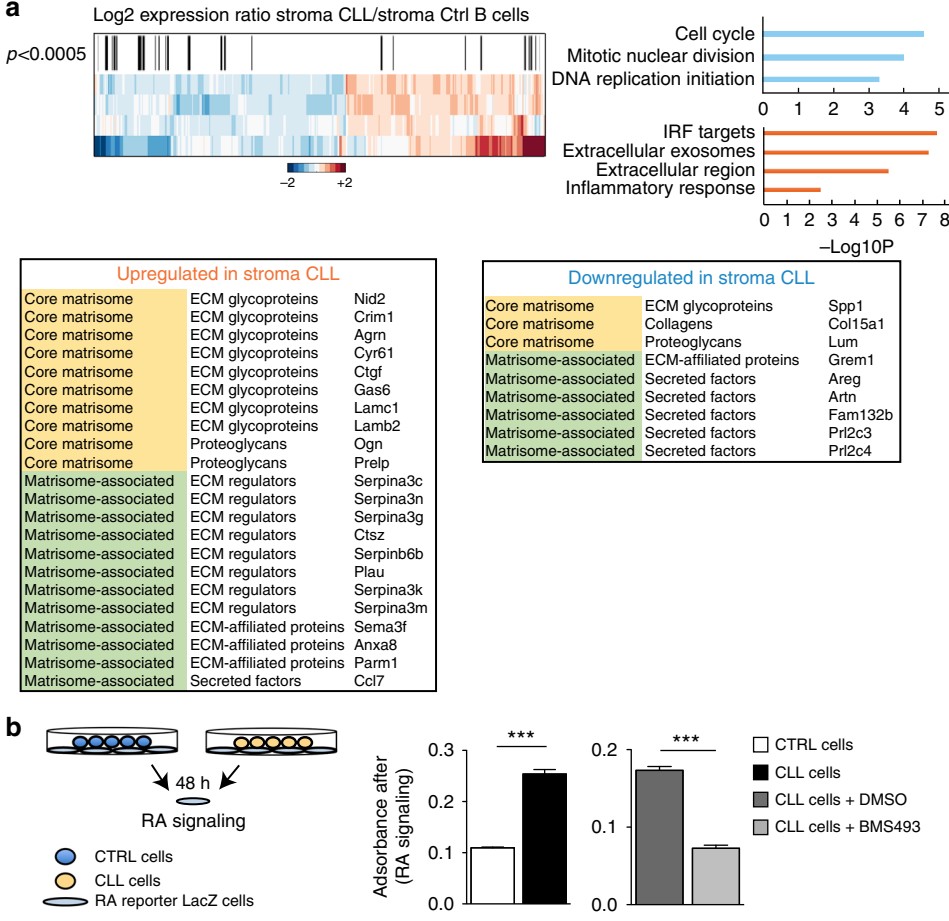

**Fig. 1** Leukemic cells induce retinoid-signaling in the stromal microenvironment. **a** Heat map of deregulated genes and their annotation analysis for the Illumina microarray. Black bars represent genes belonging to the collection of the MatrisomeDB atlas (see lists); enrichment significance was computed according to the hypergeometric distribution. **b** Experimental design of RA-reporter cells cultured with murine *Eμ-TCL1* CLL or control splenic B cells (left). RA-signaling was measured by quantifying the β-gal activity (absorbance after ONPG staining) (left) after co-cultures, and treatments with vehicle (DMSO) or with BMS493 (right). Data are representative of six independent experiments. The mean of triplicates and ±SD are shown, ***$p < 0.001$

stromal cells. To this end, we performed RNA-seq on stromal cells (mSSC line) treated with BMS493 or control vehicle (Fig. 2a). Transcriptome analysis revealed a significant down-regulation in the expression of genes involved in cellular adhesion, chemokine secretion, ECM-receptor interactions, and migration (Fig. 2a). Among the down-regulated transcripts, we validated a set of genes involved in retinoid metabolism (e.g., *Cyp26b1*, *Aldh1a1*) cellular adhesion (e.g., *Vcam-1* and *Itgα1*), migration (e.g., *Cxcl12*), and genes involved in stromal cell activation (e.g., *Acta2/αSma*) (Fig. 2a). Ablation of RA-signaling also significantly reduced the expression of genes involved in ECM remodeling, including *Loxl2*, *Lama5*, *Nidogen2*, *Col1a1*, *Col3a1*, and *Col4a6* (Fig. 2a).

Notably, we found that 30% of the genes deregulated in stromal cells after leukemia co-culture were modulated in the opposite manner following BMS493 treatment of stromal cells (Fig. 2b), confirming that leukemia modulates the expression of a large fraction of genes in stromal cells via RA nuclear receptor-signaling. Based on the transcriptomic analysis of murine stromal cells treated with BMS493, we decided to functionally validate the possibility to alter stroma–leukemia interactions and cellular adhesion through inhibition of retinoid-signaling in stromal cells. To this end, we first treated stromal cells alone with the BMS493 inhibitor for 72 h, and then added to the culture murine *Eμ-TCL1* CLL cells for the remaining 18 h in the absence of the inhibitor.

We then quantified the percentage of murine *Eμ-TCL1* CLL cells adherent to the monolayer of stromal cells. The results showed that RA-signaling inhibition reduces adhesion of murine leukemic cells to the stroma as compared to control (Fig. 2c). To further test this under conditions that more closely mimic the in vivo treatment where the inhibitor would target both leukemic and stromal cells, we set up a 3D co-culture model. For this purpose, we aggregated stromal and leukemic cells to form a spheroid in the presence of BMS493 or control vehicle (Fig. 2d). Under these conditions, we observed that while the overall number of stromal cells did not significantly change during the 3D co-culture period, the number of CLL cells that took part to the leukemic aggregate was significantly reduced in BMS493-treated spheroids (Fig. 2d). These functional data are consistent with the RNA-seq results, and indicate that retinoid-signaling strengthen stroma–leukemia interactions by promoting cellular adhesion.

**Enhanced RA activity contributes to CXCL13 expression.** As retinoids have been implicated in regulating different cellular processes and target genes, including *Cxcl13* expression[31], we hypothesized that by strengthening stroma–leukemia interactions, the enhanced RA-activity may contribute to induce *Cxcl13* in stromal cells of the microenvironment.

Gene expression profile analysis revealed a significant increase in *Cxcl13* mRNA levels in stromal cells (mSSC) cultured with leukemia as compared to control cells (Fig. 3a). Notably, treatment with the RA-signaling inhibitor BMS493 abrogated the induction of *Cxcl13* and *Rarβ*, a known target of the RA-signaling (Fig. 3a). To corroborate these findings in a more controlled setting, we cultured stromal cells in a vitamin A deficient media, either in the presence or absence of exogenous RA. Under these conditions, we found significant induction of *Cxcl13* and *Rarβ* expression by RA, which was blocked by BMS493 treatment (Fig. 3b).

We then tested whether RA inhibition can affect the distribution of CXCL13 in vivo. To this end, we assessed CXCL13 in the spleen of wild-type mice following exposure to

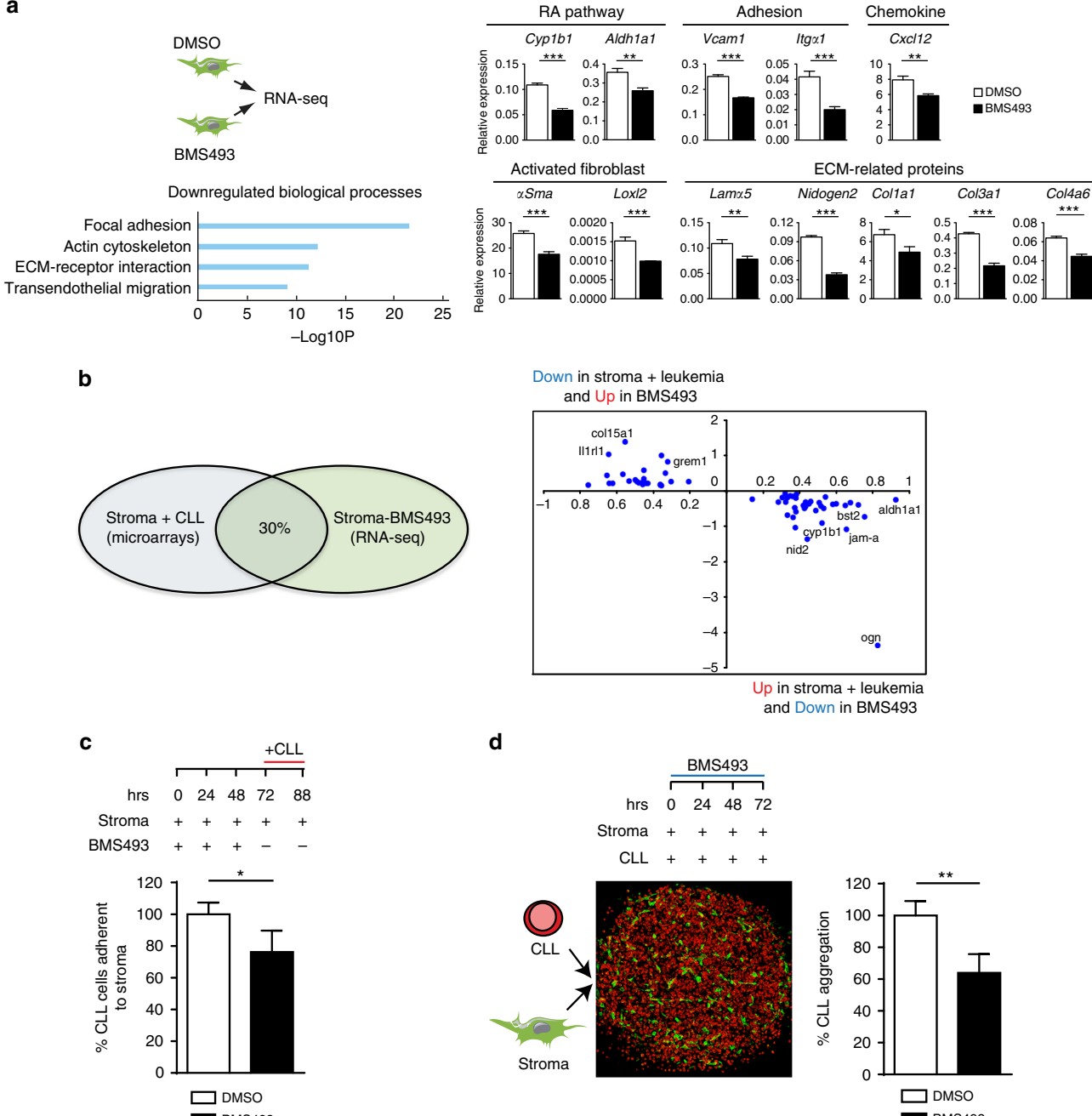

**Fig. 2** Inhibition of the RA-signaling pathway in stromal cells affects multiple biological processes. **a** Experimental design of RNA-seq analysis performed on mRNA obtained from stromal cells treated with either vehicle (DMSO) or BMS493, and most down-regulated gene-signatures (bottom). Validation of mRNA expression levels of selected stromal gene-signatures after vehicle (DMSO) or BMS493 treatment. Data are representative of four independent experiments. The mean of triplicates and ±SD are shown, *$p < 0.05$, **$p < 0.01$, ***$p < 0.001$. **b** Interpolation of gene expression profiles between microarray and RNA-seq data. **c** Quantification of murine *Eμ-TCL1* CLL cells adherence to stromal cells after BMS493 or vehicle (DMSO) treatment. Data are representative of three independent experiments with three different leukemia preparations. The mean of triplicates and ±SD are shown, *$p < 0.05$. **d** Aggregation of murine stroma (mSSC) and murine *Eμ-TCL1* CLL cells in spheroid assay. The percentage of aggregation results from the number of murine CLL cells that contribute to spheroid formation after treatment with BMS493 or vehicle (DMSO). Data are representative of one out of four independent experiments. The mean of triplicates and ±SD are shown, **$p < 0.01$

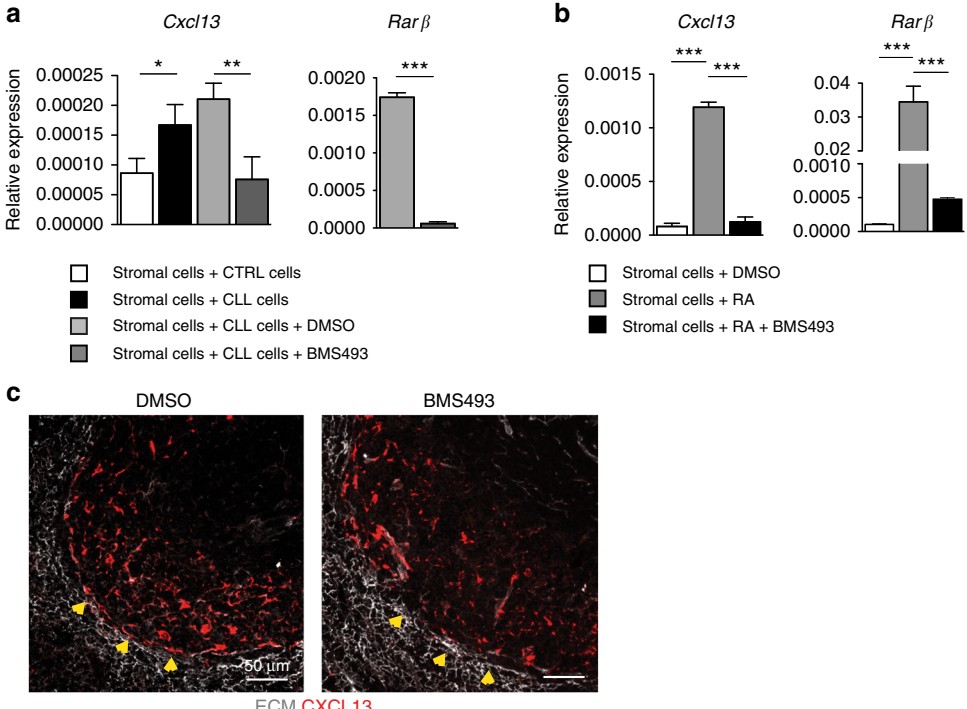

**Fig. 3** Leukemic cells induce *Cxcl13* expression in stromal cells through retinoid-signaling. **a** qPCR analysis of *Cxcl13* expression from a murine stromal cell line (mSSC) cultured with murine *Eµ-TCL1* CLL cells or control splenic B cells, and after treatment with DMSO or BMS493. Expression of *Rarβ* after BMS493 treatment was used as control. The mean of triplicates and ±SD are shown, * $p < 0.05$, ** $p < 0.01$, ***$p < 0.001$. **b** Representative qPCR analysis of *Cxcl13* expression from a murine stromal cell line (mSSC) cultured in vitamin A deficient media and treated with vehicle (DMSO), retinoic acid (RA), or BMS493. Expression of *Rarβ* after BMS493 treatment was used as control. qPCR data are representative of one out of three independent experiments. The mean of triplicates and ±SD are shown, ***$p < 0.001$. **c** Representative confocal images of the spleen from wild type mice treated with either DMSO (left) or BMS493 (right). Tissue sections were stained for laminin (ECM) (gray) and CXCL13 (red). Yellow arrowheads indicate the outer follicular region. Images are representative of one out of three mice analyzed. Scale bars = 50 μm

BMS493. The analysis revealed a consistent reduction of CXCL13 distribution particularly in the outer follicular region corresponding to the marginal reticular cell layer that also appeared more disorganized as compared to controls (Fig. 3c). Together, these findings indicate that leukemia promotes a retinoic acid-enriched microenvironment that contributes, at least in part, to CXCL13 induction.

**CXCL13$^+$ stromal cell expansion during disease progression**. Having established that leukemia promotes increased retinoid metabolism, and that RA controls CXCL13 in stromal cells, we then assessed the distribution of CXCL13 during leukemia development. To this end, we performed confocal mosaic imaging in the spleen of *Eµ-TCL1* transgenic mice or in adoptively transplanted mice with low, intermediate, and high leukemia content. The analysis revealed different patterns of CXCL13 (Fig. 4a), and showed that in a large fraction of mice (50%), the distribution of CXCL13 was increased in different leukemic contexts, and showed a predominant ring-like pattern in the outer region of the follicle corresponding to the area occupied by marginal reticular cells (Fig. 4a, pattern #1). In a fraction of transplanted mice, the CXCL13 meshwork appeared increased, and diffused throughout the entire white pulp (Fig. 4a, pattern #3). Co-localization experiments revealed that a large fraction of CXCL13$^+$ stromal cells did not express the FDC marker CD35 (Supplementary Fig. 4a). Co-staining with MOMA-1, a marker of marginal metallophilic macrophages revealed no contribution of this cell type to CXCL13 protein (Supplementary Fig. 4b).

To further assess whether murine CLL cells can induce CXCL13 in stromal cells, we injected *Eµ-TCL1* leukemic cells into *Rag2$^{-/-}$γc$^{-/-}$* mice lacking CXCL13 and mature follicular dendritic cells (FDCs)[32]. We found that, besides being capable of inducing the formation of small foci of CD35$^+$ FDC networks, leukemic cells promoted the formation of large areas of CXCL13$^+$ stromal cells not expressing mature FDC markers (Fig. 4b). Based on this, we tested whether the increase in CXCL13 signal observed during leukemogenesis may result from the proliferation of CXCL13$^+$ stromal cells, or from the induction of CXCL13 in resident stromal cells not expressing the chemokine. To this end, we took advantage of the *Pdgfra$^{gfp/+}$* knock-in mouse model in which the *Pdgfra* promoter drives the expression of nuclear green fluorescence protein (GFP)[33]. In this model, a large majority of T-cell zone fibroblastic reticular cells (FRCs) and marginal reticular cells (MRCs), and a fraction of FDCs can be visualized by nuclear GFP expression, thus allowing a more precise visualization of non-hematopoietic stromal cells. Immunofluorescence analysis performed on *Pdgfra$^{gfp/+}$* mice transplanted with *Eµ-TCL1* leukemic cells revealed that the number of CXCL13$^+$GFP$^+$ cells over the total GFP$^+$ population increased during disease progression (Fig. 4c). To discriminate between induction of CXCL13 in stromal cells previously negative for this chemokine or active stromal cell proliferation, we injected mice with 5-ethynyl-2 deoxyuridine (EdU). The analysis revealed no statistically significant differences in the proliferation of GFP$^+$ stromal cells and consistently with this, we observed that despite the increase in spleen cellularity due to leukemia infiltration, the number of GFP$^+$ stromal cells per field diminished instead of

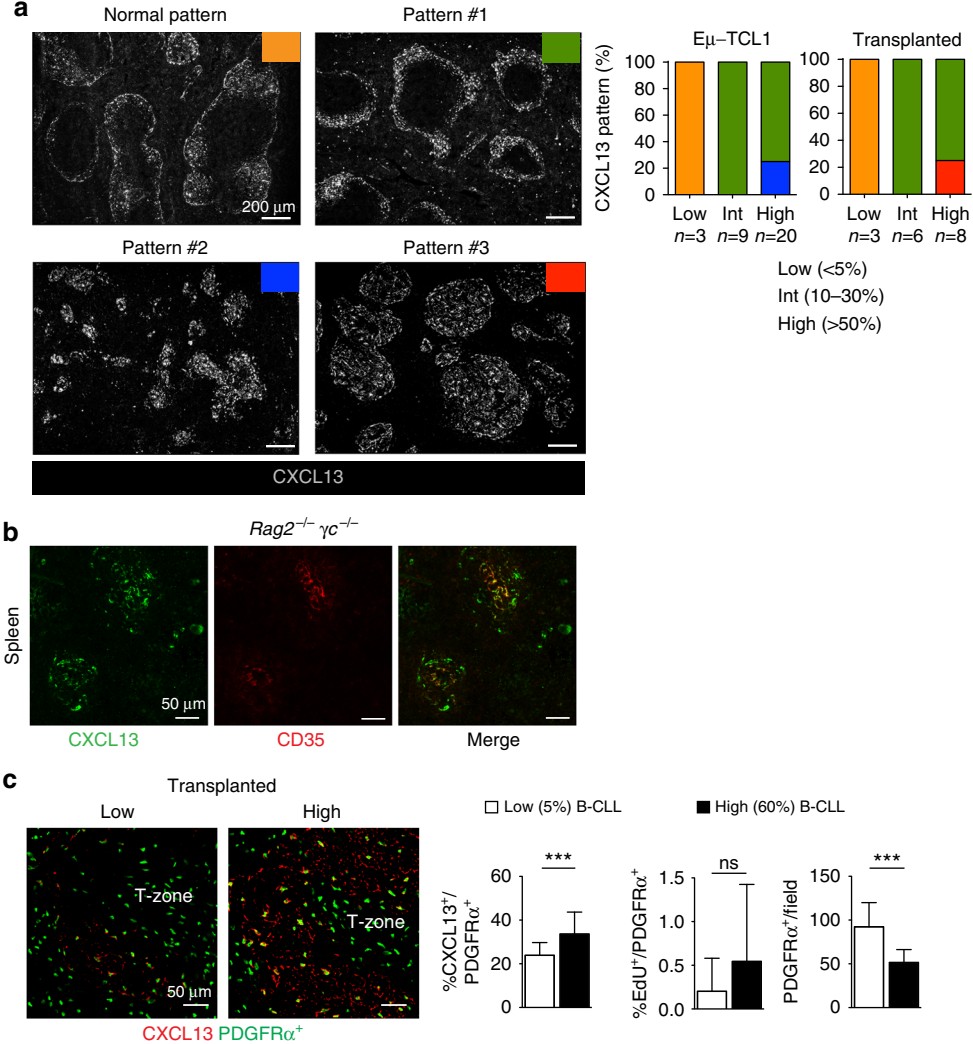

**Fig. 4** Leukemia development is associated with increased CXCL13 in stromal cells. **a** Representative confocal mosaic images of the spleen. Tissues were isolated from mice with low (<5%), intermediate (10–30%), and high (>50%) leukemia infiltration and stained for CXCL13 (gray) to visualized follicular stromal cells. Graphs indicate the frequency of the different CXCL13 patterns (normal, #1, #2, and #3). Scale bars = 200 μm. **b** Representative confocal images of spleen sections from $Rag2^{-/-}\gamma c^{-/-}$ mice injected with leukemic cells, and stained three-weeks after with CD35 (red) or CXCL13 (green) to visualize stromal cells. Each staining is representative of three mice analyzed. Scale bars = 50 μm. **c** Representative confocal images of the spleen from $Pdgfra^{gfp/+}$ mice injected with EdU to assess proliferation at different stages of leukemia development. Tissues were stained for GFP (green) to visualize PDGFRα⁺ cells and CXCL13 (red). Graphs represent: (i) the percentage of EdU⁺PDGFRα⁺ proliferating stromal cells (middle); (ii) the number of CXCL13 ⁺PDGFRα⁺ stromal cells (left); and (iii) the number (density) of PDGFRα⁺ stromal cells per field (right). Each count represents the mean ± SD of cells from seven to ten fields analyzed for each tissue. ***$p$ < 0.001. Scale bars = 50 μm. Each staining is representative of one out of three to five mice analyzed

increasing, further suggesting that CXCL13 expansion likely results from induction rather than from an increase in stromal cell proliferation. This is consistent with the microarray data, which revealed a down-regulation of genes related to cell cycle and cell division (Fig. 1a), further indicating that leukemic cells do not promote stromal cell proliferation. Altogether, these data indicate that leukemic cells induce a CXCL13-rich stromal microenvironment that favors the recruitment and accumulation of neoplastic cells in the spleen.

**Stromal cell remodeling accompanies leukemogenesis.** Given that FDCs are also involved in secreting CXCL13, we then assessed the distribution of these specialized stromal cells in *Eµ-TCL1* transgenic mice and in wild-type mice transplanted with leukemic cells, at different stages of disease progression. Confocal mosaic imaging demonstrated a progressive reduction or loss of

CD35⁺ FDCs networks in a large fraction of mice analyzed (Supplementary Fig. 5a, pattern #1 and #4). This phenotype was already evident in mice with intermediate (10–30%) percentage of leukemia in the peripheral blood, and it was maintained in the majority of mice with high (≥50%) leukemia. In a fraction (35%) of mice, however, we found persistence or expansion of disorganized CD35⁺ FDCs clusters (Supplementary Fig. 5a pattern # 2 and #3). Altogether, these findings further demonstrate that leukemic cells promote CXCL13 in follicular stromal cells different from conventional FDCs.

We then evaluated the pattern of follicular stromal cells during the course of human CLL. We stained human CLL splenic biopsies with CD21 and MAdCAM-1 to visualize FDCs and MRCs, respectively, which are known to express CXCL13. The analysis revealed a disorganized pattern of FDCs and MRCs, and the presence of CXCL13⁺ reticular stromal cells in all CLL spleens tested (Supplementary Fig. 5b and c).

Immunohistochemistry analysis also revealed the presence of CXCL13 in lymph node (LN) biopsies of CLL patients, and a significant increase of CXCL13⁺ cells during disease transformation (Supplementary Fig. 5d). Together, these findings demonstrate that CLL cells induce remodeling of the splenic follicular stromal compartment, and reveal that CXCL13 is modulated during the evolution of murine and human CLL.

**Targeting RA-signaling prolongs leukemia survival.** Taken together our data suggest that inhibition of the RA-signaling could affect disease progression by altering the stroma/ECM–leukemia interactions and the chemokine networks. Thus, to test the possibility that inhibition of this signaling pathway may impact murine CLL progression, we exploited several approaches to block RA signaling in vivo. First, we generated mice deficient in vitamin A (VAD mice) from the day of birth[34] and transplanted them at 2 months of age with Eµ-TCL1 leukemic cells. We observed that leukemia engraftment was strongly suppressed in the spleen and bone marrow (BM) of VAD mice, as compared to mice fed with a control diet (Fig. 5a). We then established a more physiological model, in which retinoids were gradually depleted over time. To this end, we fed 2 months old Eµ-TCL1 mice with a vitamin A deficient (VAD) or control diet before leukemia onset. Long-term survival analysis revealed that VAD Eµ-TCL1 mice survived longer (Fig. 5b), and this corresponded to significantly reduced levels of retinoic acid precursors as compared to control mice (Fig. 5b).

We then aimed to establish the potential of retinoid-antagonist therapy on leukemia progression. We first treated wild-type mice with the BMS493 or control vehicle 1 week after the injection of $1 \times 10^7$ leukemic cells. We found that inhibition of RA-signaling significantly delayed leukemia onset and reduced tumor expansion (Fig. 5c). In a separate cohort of mice analyzed, we found that treatment diminished the accumulation of leukemic cells in the spleen, bone marrow, peritoneal cavity, and peripheral blood (Fig. 5c). Notably, annexin-V and Ki67 staining did not reveal differences in apoptosis or proliferation, respectively, in mice treated as compared to controls (data not shown). Next, we performed a survival curve upon BMS493 treatment. The results show that wild-type mice engrafted with Eµ-TCL1 CLL cells and treated with BMS493 ($n = 10$) survive significantly longer as compared to controls (Fig. 5d). Consistent with this, echographical measurement of the spleen size during disease progression (day 35) revealed that at this stage treatment with BMS493 significantly reduced infiltration of leukemic cells in the spleen (Fig. 5d). These findings demonstrate that blocking retinoid-signaling may represent an effective strategy to control murine CLL progression.

**Blocking RA-signaling controls peritoneal leukemia expansion.** Fat-associated lymphoid clusters (FALCs) are atypical lymphoid tissues of the peritoneal cavity that are induced by inflammation and contain different cell subsets including CXCL13⁺ non-hematopoietic stromal cells[35,36]. Given that leukemic cells from Eµ-TCL1 mice share features of peritoneal B-1 cells, a cell type which is also present in the FALC, we hypothesized that peritoneal FALCs may represent supportive niches for murine CLL expansion. To test this, we first assessed the homing of leukemic cells to the omental FALCs by injecting green-labeled Eµ-TCL1 CLL cells into the tail vein of wild-type recipients. Immuno-fluorescence confocal analysis performed 48 h post injection revealed a substantial accumulation of murine CLL cells in this site (Supplementary Fig. 6a). We then analyzed the peritoneal cavity of Eµ-TCL1 transgenic mice with a high percentage of leukemia cells in their peripheral blood. In these mice, we found

enlarged omental FALCs, and an increased number of mesenteric FALCs containing a high percentage of CD19⁺CD5⁺ leukemic cells as compared to control mice in which mesenteric FALCs were barely detected (Supplementary Fig. 6b and c). Moreover, FALCs from leukemic mice were significantly larger and characterized by a considerable expansion of CXCL13⁺ cells distributed within a network of collagen-IV (Supplementary Fig. 6b and c).

Interestingly, gross morphology analysis revealed that inhibition of RA-signaling significantly suppressed induction of mesenteric FALCs in transplanted mice (Supplementary Fig. 6d). Altogether, these findings demonstrate that peritoneal FALCs are supportive niches for the growth of leukemia, and that antagonizing retinoid-signaling controls FALC formation and disease expansion in the peritoneum.

**Increased expression of RA-nuclear receptors in human CLL.** Previous work showed that RA induces the expression of different target genes including RA-nuclear receptors[37]. Based on these findings, we hypothesized that increased RA activity within the microenvironment may promote induction of RA-associated genes in responder leukemic cells. To test this hypothesis, we first assessed the expression of genes belonging to the RA pathway in CLL cells freshly isolated from the spleen of Eµ-TCL1 mice. Among different genes tested, we found that expression of Rary2 was significantly increased in leukemia as compared to control splenic B cells (Fig. 6a).

We then analyzed expression of RA-nuclear receptors in a previously published human data set of CLL patients[38], and found that human CLL cells isolated from the peripheral blood express higher levels of RXRα as compared to normal B cells (Fig. 6b, left panel). Furthermore, we screened 60 human primary CLL cases with different genomic aberrations, and found a significant increase in RXRα expression in CLL cells of patients with 17p and/or 11q deletions and worse prognosis, as compared to those with 13q deletion and better outcome (Fig. 6b, right panel). Moreover, the RXRα expression is independent from other parameters such as immunoglobulin heavy chain variable region (IGHV) gene mutational status, CD38 and ZAP70 expression (Fig. 6c). These data demonstrate that RA nuclear receptors are up-regulated in human CLL cells, and that their expression levels identify a subset of patients with bad prognosis; and that human CLL cells are equipped to respond to endogenous retinoids.

## Discussion

In mouse and human CLL, the crosstalk between leukemia and the surrounding microenvironment promotes tumor cell survival and disease progression[2,3].

Consistent with the notion that leukemic cells modify the microenvironment, gene expression profile (GEP) analysis of stromal cells cultured with murine Eµ-TCL1 CLL cells revealed deregulation of genes involved in inflammation and stroma/extracellular matrix remodeling. Interestingly, our findings revealed that human leukemic cells also deregulate similar gene signatures, thus indicating commonalities between mouse and human CLL cells. Among those genes, we found up-regulation of αSma/Acta2, a marker of activated and cancer-associated fibroblasts, and Nidogen-2, Lamb2, and Prelp, which encode for ECM-associated proteins. In addition, PRELP and its family member FMOD are two ECM secreted glycoproteins overexpressed in CLL cells, and whose role in CLL remains unknown[39,40]. The analysis also revealed that mouse leukemic cells, and to some extent human CLL cell as well promote retinoid-synthesis and signaling, and that retinoids contribute to tissue remodeling and disease progression in mouse

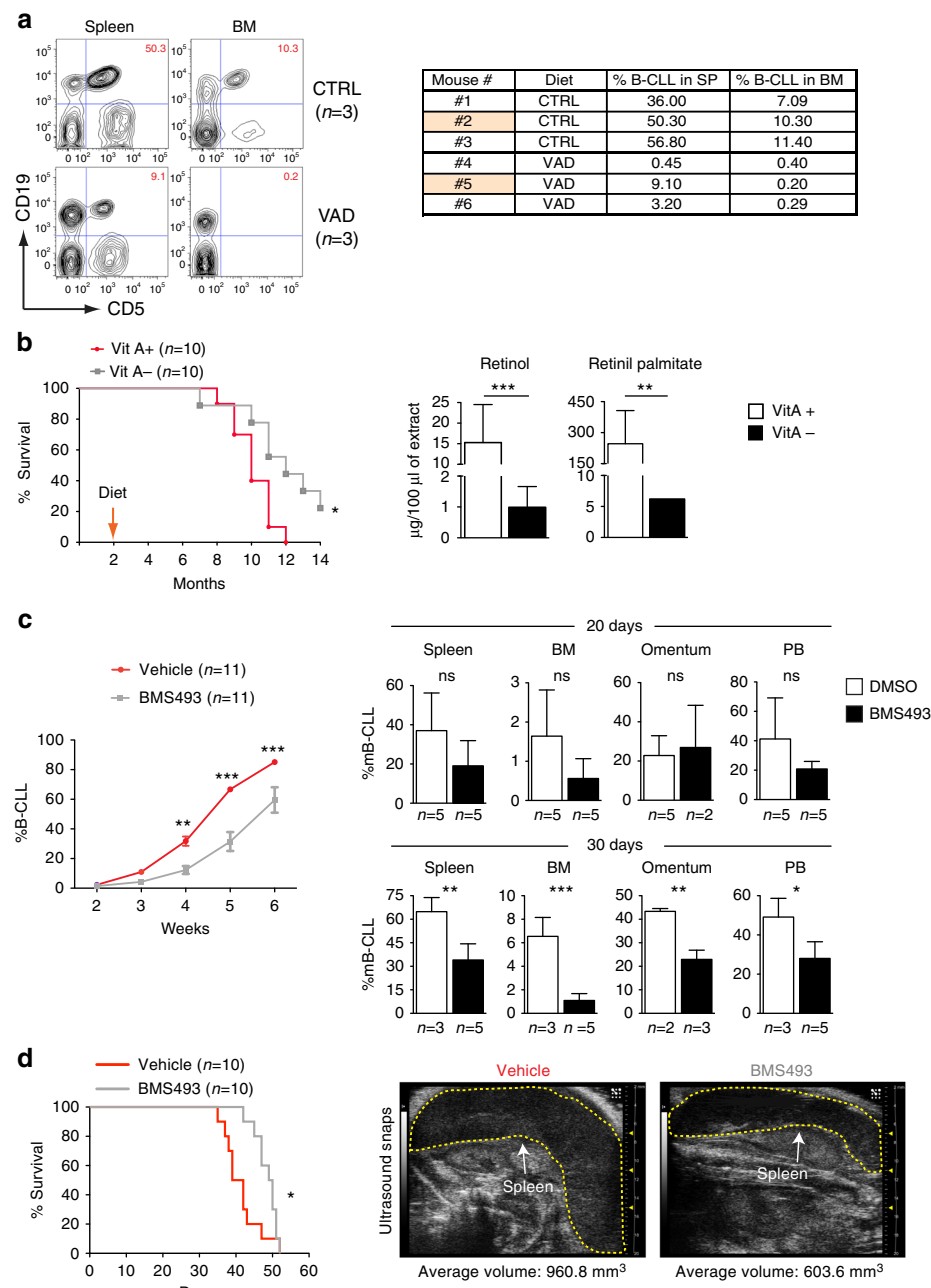

**Fig. 5** Targeting RA-signaling controls disease progression and prolongs survival. **a** FACS analysis of control mice ($n = 3$) or VAD ($n = 3$) injected with leukemic cells. The table summarizes the percentage of murine $E\mu$-TCL1 CLL cells in spleen and BM for each mice analyzed. **b** Survival curve for $E\mu$-TCL1 mice fed with vitamin A deficient (VitA−) or control (VitA+) diet. Quantification of RA-precursors retinol and retinyl palmitate in the liver of moribund $E\mu$-TCL1 VitA+ or VitA− fed mice. The mean of triplicates and ±SD are shown, *$p < 0.05$, **$p < 0.01$ and ***$p < 0.001$. **c** Analysis of leukemia progression over time in wild type mice transplanted with leukemic cells and treated with vehicle (DMSO) or BMS493. Flow cytometry analysis of leukemia cell infiltration in different tissues at 20 or 30 days after the indicated treatment, *$p < 0.05$, **$p < 0.01$ and ***$p < 0.001$. **d** Survival curve of wild-type mice transplanted with $E\mu$-TCL1 CLL cells and treated with vehicle (DMSO) or BMS493, *$p < 0.05$. Representative ultrasound images and echographical measurement of the spleen size performed at day 35 of disease progression

models. Indeed, a murine stromal cell line treated with the RA-signaling inhibitor showed repression of genes involved in adhesion, ECM–cell interactions, migration and tissue remodeling. Interestingly, our findings indicate that 30% of the genes deregulated in the stromal cell line after leukemia co-culture are controlled by RA nuclear receptors, thus indicating that activation of retinoid signaling upon stroma–leukemia interactions are responsible for regulating a large subset of stromal-associated genes including genes encoding for chemokines.

Our work in mice demonstrates that the CLL-dependent induction of *Cxcl13* in stromal cells is, at least in part, retinoid-dependent, since treatment with the RA antagonist BMS493 prevents *Cxcl13* induction. In line with this, treatment of spleen stromal cells with RA induces *Cxcl13* expression. Consistent with our results and with previous findings indicating that retinoids regulate *Cxcl13* expression[31,41], inhibition of RA-signaling in mice causes a slight but consistent reduction and disorganization of spleen CXCL13 distribution.

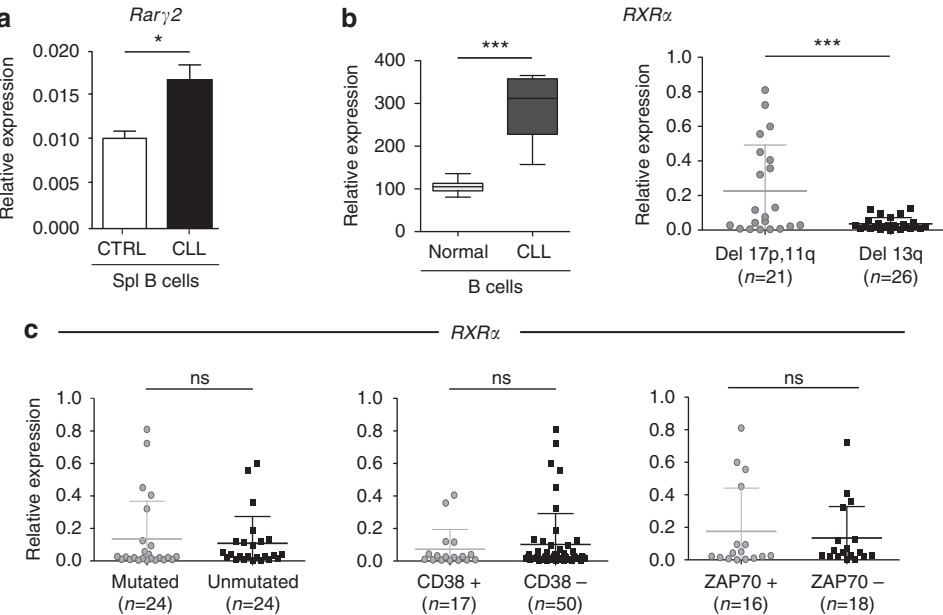

**Fig. 6** Increased expression of RA-nuclear receptors in human primary CLL cells correlates with bad prognosis. **a** qPCR analysis of *Rarγ2* expression in *Eμ-TCL1* CLL and control B cells. Data are representative of one out of five mice analyzed. The mean of triplicates and ±SD are shown. *$p < 0.05$. **b** Relative *RXRα* expression data of purified B-cells from healthy donors and CLL patients (**b**, left) from public repository, and expression of *RXRα* in CLL patients with different genomic aberrations (**b**, right). ***$p < 0.001$. **c** Correlation analysis of RXRα expression from purified CLL cells from patients with different IGHV mutational status, CD38 and ZAP70 expression

In agreement with our in vitro findings, the distribution of CXCL13 also increased during disease progression. This was particularly evident in transplanted mice, in which CXCL13+ stromal cells are spread throughout the white pulp, and in the outer follicular region corresponding to the marginal reticular cell layer where they form a thicker ring of stromal cells positive for this chemokine. Although we found that CXCL13 was also expressed by FDCs, our results indicate that to a major extent follicular stromal cells different from FDCs express CXCL13 during leukemia progression. This is further supported by the fact that the signal of CXCL13 remained abundant in stromal cells not expressing FDC markers. Our findings revealed that in mice, the expansion of CXCL13+ cells result from induction of CXCL13 in stromal cells previously negative for this chemokine, rather than an increase in stromal proliferation. However, we cannot exclude the possibility that a slow-rate proliferation of CXCL13+ stromal cells takes place during disease progression and may contribute to CXCL13 expansion. Of note, recent work showed that during inflammation, B-cell follicles expanding into the T-cell zone convert pre-existing stromal cells, positioned at the B-T border, into CXCL13+ secreting cells[42]. These cells, different from conventional T-zone FRCs, were termed versatile stromal cells. A similar mechanism might operate during murine CLL progression and contribute to observed phenomena, where leukemic cells expanding into the T-cell zone change the phenotype of local stromal cells.

Similar to the mouse model, we found altered stromal composition and loss of FDCs in the spleen of CLL patients. We also observed the presence of CXCL13 in human CLL spleen and lymph nodes, and that the number of CXCL13+ cells increases during disease transformation. Our findings demonstrate that leukemia progression induces changes in the follicular stromal microenvironment, and indicate that FDCs are dispensable for the late phase of CLL progression, and likely play a role only in the early phase, as previously proposed[25].

Based on the GEP data, we hypothesized that preventing RA-signaling could be a strategy to interfere with stroma–leukemia

interactions at multiple levels. Indeed, our results showed that mice fed with a VAD diet or treated with an RA-signaling inhibitor survive longer as compared to controls, and this phenotype appears to be the result of reduced accumulation and expansion of leukemic cells in lymphoid tissues. Although our data point to a role of retinoids in modulating stroma–leukemia interactions at multiple levels, we cannot exclude that RA-signaling inhibition may also have a cell autonomous effect on leukemic cells. This hypothesis is based on findings showing that RA-signaling plays a role in the development of marginal zone B cells and B1 cells[43].

Our findings also reveal that mesenteric FALCs represent additional leukemia-supportive niches, and that retinoid-antagonist therapy prevents FALC formation, and consequently the expansion of leukemia in the peritoneal cavity. These findings are in line with the notion that RA was shown to affect the migration of B cells to the gut by inducing α4β7 integrin[43]. Furthermore, given the high retinoids content of the peritoneal adipose tissue, it is also likely that RA-signaling impacts on *Cxcl13* expression, homing of leukemia to the peritoneal cavity, and consequently FALC formation. Of note, peritoneal infiltration and enlargement of abdominal LNs occurs in CLL and other blood malignancies[44]. Although several reports have linked omental FALCs with solid tumor metastasis, it is unknown whether FALCs are induced in human CLL, and function as supportive niches promoting disease progression.

Based on our findings in mice, we propose that leukemic cells are capable to increase retinoic acid activity in the microenvironment. In addition, our work also indicates that upon interaction with mouse and human leukemic cells, stromal cells can up-regulate the enzymes involved in RA synthesis, which can act in an autocrine and paracrine fashion to stimulate multiple pathways, including ECM–leukemia interactions, adhesion, and chemokine secretion, and possibly in combination with other pathways such as toll-like receptor signaling, as previously demonstrated[45]. In addition, we suggest that RA acting in a paracrine manner activates target genes in both neoplastic and other non-tumoral cells of the microenvironment. Recently, a

retinoic acid-low microenvironment in multiple myeloma (MM) was shown to prevent differentiation of MM cells and promote drug resistance[46], a mechanism that we have not investigated. However, previous work indicate that retinoic acid does not promote differentiation of CLL cells[47].

Although future work is required to fully elucidate the role of RA in the CLL microenvironment, our findings reveal that mouse and human leukemic cells could be distinguished from normal B-cells by their increased expression of *Rarγ2* and *RXRα* respectively, thus indicating they are equipped with the machinery to activate RA-target genes upon RA-binding. At present, the meaning of the correlation between increase expression of *RXRα* in a subset of patients carrying high-risk genetic aberrations and bad prognosis remains unclear. Nevertheless, emerging data point to a critical contribution of *RXRα* in lymphoid malignancies[48]. In support of this, loss of *Rxrα* protects mice from developing leukemia[49]. It is worth mentioning that RXRα is a dimerization partner for other nuclear receptors such LXRs, VDRs, and PPARs, and thus it might be involved in the regulation of other signaling pathways implicated in CLL pathogenesis.

In cancer, agonists of RA nuclear receptors have been used with the rational to induce terminal differentiation, inhibit proliferation and promote apoptosis[20,21]. Interestingly, using in vitro models it was demonstrated that both agonists and antagonists of RA-signaling induce similar growth inhibitory effects in cancer cells, thus indicating that inhibition of RA signaling is also detrimental and may represent a strategy for cancer treatment as we propose[50].

In conclusion, although our findings were mostly obtained using mouse models that may not fully reflect the human CLL, they indicate that retinoid-signaling plays an important role in the pathogenesis of murine CLL, and that retinoid-antagonist therapy may represent an effective strategy to target CLL-microenvironmental interactions at multiple levels to control disease progression.

## Methods

**Mice**. C57BL6 were purchased from Charles River Italia; *Eμ-TCL1*, *Rag2*$^{-/-}γ_c^{-/-}$[48–51] and *Pdgfrα*$^{gfp/+}$ mice have been previously described[33,51,52]. For in vivo analysis, female and male mice of 8–10 weeks of age were used. Animals were maintained in a specific pathogen-free animal facility and treated in accordance with European Union and Institutional Animal Care and Use Committee guidelines.

**Immunofluorescence staining and confocal analysis**. Tissues were collected and fixed for 5 min in 4% (w/v) PFA (Sigma-Aldrich), then washed in PBS 1 × and dehydrated overnight in 30% sucrose (Sigma-Aldrich) at 4 °C. Samples were embedded in Tissue-Tek OCT compound (Bio-optica) and frozen in ethanol dry-ice bath (using Dehyol 95 (Bio-optica)). Around 8–10 μm thick sections were placed onto glass slides (Bio-optica), fixed in cold acetone for 5 min, dried, and kept at −80 °C until used. Slides were incubated 30 min with a blocking solution of PBS at 0.5%FBS and 0.05% Tween (VWR) (PBS-T 0.05%), followed by primary (supplementary material) specific antibodies or secondary reagents (supplementary material). Primary antibodies, secondary antibodies, and streptavidin reagents were diluted in blocking solution PBS-T 0.05% and incubated for 1 h and 30 min, respectively. For anti-mouse and biotin-conjugated primary antibodies, additional incubation with MoM (Vector Lab) and Avidin/Biotin blocking solution (Vector Lab), respectively, were performed following the manufacturer's instructions. Nuclei were visualized with DAPI (Fluka), and mounting was performed with Mowiol (Calbiochem). For detection of MAdCAM-1, CXCL13 and PDPN antibodies, Tyramide Signal Amplification kit (Perkin Elmer) was used. To visualize proliferating cells, Click-iT® EdU Imaging Kits (Invitrogen) was employed accordingly to manufacturer's protocol. Confocal images were acquired using Leica TCS SP2 and Leica TCS SP8 microscopes. Digital images were recorded in separately scanned channels with no overlap in detection of emissions from the respective fluorochromes. Final image processing was performed with Adobe Illustrator CS4 and Adobe Photoshop CS4. For immunohistochemistry of human biopsies, tissue sections from formalin-fixed, paraffin-embedded blocks were used. Sections were stained with primary goat polyclonal antibody to CXCL13 (dilution 1:30; R&D Systems) and upon appropriate antigen retrieval, reactivity was revealed using biotinilated anti-goat IgG (dilution 1:250; Vector Laboratories) followed by 3,3′-diaminobenizidine tetrahydrochloride (DAB), and finally counterstained with H&E according to standard protocols. After dehydratation slides were permanently

mounted in Eukitt (Bioγ-Optica). Digital images were acquired using the Olympus BX60 microscope with DP-70 Olympus digital camera and processed using Analysis Image Processing software.

**Isolation, purification, and characterization of leukemic cells**. *Eμ-TCL1*, transplanted and control mouse tissues (peripheral blood, lymph nodes, omentum, mesentery, and femoral bone marrow) were collected from mice either alive (peripheral blood) or killed (other tissues). Solid tissues were smashed and filtered on a 40 μm cell strainer (Corning). Single-cell suspensions were washed in PBS and erythrocytes were depleted using an ammonium chloride solution (ACK) lysis buffer (NH₄Cl 0.15 M, KHCO₃ 10 mM, Na₂EDTA 0.1 mM, pH 7.4). For flow cytometry analyses, samples were incubated for 15 min at RT with Mouse BD Fc Block™ (purified rat anti-mouse CD16/CD32; BD Bioscience Pharmingen). Then cells were washed and incubated for 15 min at 4 °C with conjugated antibodies and finally data were acquired on FACSCanto™ II (BD Biosciences) and analyzed using FlowJo software (Tree Star). For culture experiments, B-lymphocytes were collected from the spleen, purified and enriched by negative depletion (EasySep™ Mouse B Cell Enrichment Kit; StemCell Technologies). The purity of all murine CLL samples and control B cells was always more than 90%. Human CLL cells were purified immediately after blood withdrawal, by negative depletion using the RosetteSep B-lymphocyte enrichment kit (StemCell Technologies). The purity of all human preparations was always more than 99% and the cells co-expressed CD19 and CD5 on their cell surface as checked by flow cytometry (FC500; Beckman Coulter); preparations were virtually devoid of natural killer cells, T lymphocytes, and monocytes. Human primary samples were obtained from RAI stage 0-1 CLL patients, after informed consent as approved by the Institutional committee (protocol ViVi-CLL) of San Raffaele Scientific Institute (Milan, Italy) in accordance with the Declaration of Helsinki.

**Cell cultures and treatments**. A mouse spleen-derived stromal cell line (mSSC) expressing *yfp* was generated as previously described[4]. F9-RARE-LacZ cell line was previously described[37]. Cells were cultured at 37 °C, 5% CO₂ in DMEM (Gibco) supplemented with 10% heat-inactivated FBS (Euroclone), 2 mM L-glutamine (L-Glu; Gibco), 100 U/ml penicillin and 100 μg/ml streptomycin (Pen/Strep; Gibco). All cell lines are monthly tested for mycoplasma. For experiments in Vitamin A deficient medium, the FBS serum was substituted with B27 supplement normal and without retinyl acetate (B27-normal and B27-vitA⁻; Invitrogen). Stromal cells were treated for 24 or 48 h with different stimuli: 1 μM all-*trans* Retinoic Acid (in DMSO; Sigma-Aldrich), 1 μM BMS493 (in DMSO; Tocris Bioscience)[30] or vehicle (DMSO).

**Leukemia propagation, co-culture assays and lymphoid aggregate formation**. For leukemia propagation, *Eμ-TCL1* mice were killed when CD19⁺CD5⁺ cells reached 90% in PB. A total of 1 × 10⁷ leukemic cells (from different donor mice) were purified form the spleen and injected intraperitoneally into syngeneic C57BL6/N, *Rag2*$^{-/-}γ_c^{-/-}$ or *Pdgfrα*$^{gfp/+}$ recipients. Treatment with the BMS493 inhibitor was performed by oral gavage at 5 mg/Kg dosage (in corn oil) three times a week for 6–8 consecutive weeks. For ex vivo organ cultures, animals were killed when terminally sick. Labeling of B-lymphocytes with CellTracker™ Green CMFDA Dye (ThermoFisher Scientific) was performed by incubating 1 × 10⁷ cells /mL for 20 min at 37 °C according to manufacture instructions. Labeled cells were injected intraperitoneally and mice killed at the indicated time points. Vitamin A Deficient mice were generated as previously described[34].

Co-culture experiments were performed using purified B-lymphocytes in combination with either mSSC or F9-RARE LacZ reporter cell line. For the co-culture experiments with F9-RARE LacZ reporter cell line, 1 × 10⁷ B-lymphocytes were seeded on top of 2.5 × 10⁵ of F9-RARE LacZ cells in poly-L-Lysine-coated 48-well plate (Costar). After 48 h, floating cells were discarded while remaining adherent cells were lysed and β-gal activity assay was assessed. Co-culture experiments were conducted in triplicate, and cells were maintained in complete DMEM with 5% CO₂ at 37 °C. Assessment of βgal activity was performed as previously described[37].

For lymphoid aggregate formation, immortalized mSSC and leukemic B-lymphocytes were used. Specifically, mSSC and *Eμ-TCL1* CLL cells were mixed in a ratio 1:20 (specifically 4 × 10⁵ cells and 80 × 10⁵ cells for each aggregate), pelleted and re-suspended in 1 ml of 0.9% type I rat tail collagen solution. This solution was prepared, on ice, with the following components: 360 μl of DMEM, 14 μl of PBS ×10, 3 μl of NaOH 1 M, and 125 μl of Collagen I (Corning), and was used immediately after the preparation. A volume of 5 μl drops of the resulting cell suspension were spotted on the lid of a petri dish, as hanging drops, and were incubated for 20–25 min in humidified incubator with 5% CO₂ at 37 °C, to favor collagen polymerization. Next, the polymerized cells-collagen drops were transferred into a petri dish with 10 ml of DMEM. After 24 h, the organoids were formed by collagen contraction[53]. For BMS493 treatment experiments: mSSCs and murine *Eμ-TCL1* leukemic cells were pre-treated individually before the organoids formation for 24 h with 1 μM of BMS493 (in DMSO; Sigma) and vehicle (DMSO). On established organoids, BMS453 or vehicle treatment was repeated for additional 48 h, with one administration per day. At the end of the treatments, pools of four organoids were digested in 60 μl of 0.225 mg/ml Liberase (Roche) solution for 10 min at 37 °C,

agitating at 1000 rpm. Digested organoids were analyzed by MACSQuant Analizer 10 (Miltenyi Biotec) in order to assess the number of live cells (DAPI negative). Further analysis was performed using FlowJo software (Tree Star).

**Microarray, RNA-seq, and qPCR analyses.** For microarray analysis, $2 \times 10^7$ purified B-lymphocytes were seeded on top of $7.5 \times 10^5$ mSSC in 60 mm dishes (Costar). After 40–48 h, floating B-lymphocytes were collected with the supernatant by gently flushing, instead $yfp^+$ stromal cells and stroma-adherent B lymphocytes were separated by sorting using MoFlo™ XDP (Beckman Coulter) cell sorter. Total RNA was extracted with RNeasy Micro or Mini kit (Qiagen). For microarray analysis, cRNA preparation and amplification was performed using Illumina TotalPrep™ RNA Amplification Kit, and then the analysis was performed using Illumina Whole-Genome Gene Expression Direct Hybridization Assay system (Illumina). For RNA-seq analysis, quality of the RNA was checked with Agilent RNA 6000 Nano chip, and run on Bioanalyzer 2100 (Agilent) (The Center of Bioinformatics and Functional Genomics at OSR performed RNA sequencing). Briefly, library preparation was performed using the Illumina TrueSeq Stranded mRNA kit (Illumina), starting from 300 ng of total RNA. After barcoding, the RNA libraries were pooled, denatured, and diluted to an 8 pM final concentration. Cluster formation was performed on cBot (Illumina) using flow cell v.3. The sequencing by synthesis (SBS) was performed according to TruSeq SR protocol (Illumina) for the HiSeq 2500 (Illumina) set to 100 cycles, yielding an average of $18 \times 10^6$ clusters for each sample. For qPCR analysis, reverse transcription of 0.2–2 µg of total RNA was performed with the ImProm-II Reverse Transcription System kit with random primers (Promega). qPCRs were performed using Universal Probe Library system (Roche) on a LightCycler480 (Roche). The $C_t$ of $Rpl13$ or $GAPDH$ (housekeeping genes for mouse and human, respectively) was subtracted from the $C_t$ of the target gene, and the relative expression was calculated as $2^{-\Delta Ct}$. qPCRs were performed in triplicate or quadruplicates and mean ± SD represented as relative expression (primer sequences are described in supplementary material).

**HPLC/UV analysis of retinyl ester (RE) and retinol (ROL).** Vitamin A depletion was assessed by measurement of retinol and retynil palmitate with UV-HPLC as previously described[54]. Solvents and controls were purchased from Sigma-Aldrich. 300–500 mg of tissue were frozen immediately after collecting and were kept at −80 °C until assay. Samples were extracted as previously described with some modifications. Immediately before the analysis, 5 µl of internal standard (IS, 20 µM retinyl acetate) were added to each sample. Then tissues were homogenized on ice using a Heavy Duo Stirrer motorized potter in 1.5 ml of cold 0.9% NaCl. A volume of 1.5 ml of 0.025 M KOH in ethanol was added to tissue homogenates and mixed at 1100 rpm, RT for 30 min, and after 5 ml hexane was added to the aqueous ethanol phase. The samples were vortexed and centrifuged for 10 min at 1200 rpm. The hexane phase containing ROL and RE was recovered. The extraction was repeated twice, the hexane phase collected and dried under nitrogen. ROL/RE extracts were resuspended in 500 µl acetonitrile. A volume of 100 µl portions were analyzed by high pressure liquid chromatography (HPLC). Separations were obtained on a LiChrospher 100 RP18 column (5 µm, 250 × 4 mm; Merck) and were quantified by UV absorbance at 325 nm. Elution was carried out at a flow rate of 1 mL/min, with gradient formed by the solvent A, consisting of water, solvent B, consisting of acetonitrile, and solvent C consisting of acetonitrile with 0.1% dichloromethane. The gradient elution program was as follows: 19 min 30% A and 70% B, 11 min linear gradient to 11% A and 89% B, 1 min 11% A and 89% B, 1 min linear gradient to 100% B, 8 min to 100%, 2 min linear gradient to 100% C, 8 min linear gradient to 100% B, and 5 min linear gradient 30% A and 70% B.

**Ultrasound imaging of the spleen.** Murine spleens were imaged using the Vevo 2100® High-Resolution Ultrasound Imaging System (VisualSonics, FUJIFILM, Toronto, Canada). 2D Ultrasound images were performed in B-mode using the Vevo 2100 linear array transducers MS 550D (center frequency of 40 MHz) and MS 250 (center frequency of 21 MHz). After removal of the fur from the abdomen of the mice using depilatory cream (Veet), a warmed ultrasound Gel was placed between the skin surface and the transducer. The sonograms were analyzed and the transversal and sagittal diameters of the widest part of the spleen, and the length of the spleen were determined using the image analysis software Vevo LAB (Visualsonic, FUJIFILM). All the ultrasonographic evaluations were performed by a single examiner.

**Statistical analysis.** Illumina microarray data were processed in the R environment. Normalization was obtained with the lumi package[55], and differential analysis was performed by applying a permutation-based non-parametric method implemented in the RankProd package[56]. Differential expressed genes were selected according to the threshold FDR $q$-value < 0.05.

Statistical analysis using a 2-tailed unpaired Student's $t$ test or 2-way Anova or Log-rank (Mantel-Cox for survival curves) test was performed with GraphPad Prism 5.0c (GraphPad Software), and values were expressed as mean ± SD or SEM (indicated in the figure legends). Differences were considered statistically significant at $p$ less than 0.05. For RNA-seq analysis, raw sequences (fastq) were filtered for good quality scores using FastQC software. FastQC: a quality control tool for high throughput sequence data. Available online at: http://www.

bioinformatics.babraham.ac.uk/projects/fastqc. Sequences obtained were aligned to the Mouse genome (mm10 release) using STAR aligner (version 2.3.0e_r291)[57]. Only uniquely mapped reads were used to estimate gene counts using the reported Ensembl gene annotations (v85) using Rsubread Bioconductor package. Subsequent to mapping the gene count, data were normalized using the "weighted trimmed mean of M-values" described elsewhere[57]. After normalization, differential gene expression was performed using the limi package in R[55]. Annotation analysis was performed using VLAD – Visual Annotation Tool at MGI (Mouse Genome Informatics)[58]. No statistical method for sample size choice was used; however, a similar number of mice is generally used in the field for similar experiments. No animals have been excluded from the analysis, and no randomization methods were used to allocate mice to different groups. For assessing experiment outcome no blinding method was used.

**Study approval.** Biological samples were obtained from patients affected by CLL according to the research protocol "ViViCLL" approved by the institutional review board at San Raffaele Hospital (Milan, Italy). Informed consent was obtained in accordance with the Declaration of Helsinki.

**Data availability.** Microarray and RNA-seq data were deposited in the Gene Expression Omnibus (GSE111613). The authors declare that all data supporting the findings of this study are available within the manuscript or its supplementary files or are available from the corresponding author upon reasonable request.

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

## Acknowledgements

The authors are grateful to lab members for technical assistance and helpful suggestions. We are thankful to Michael Wagner for the F9 RARE-LacZ reporter cell line. This work was supported by grants from the Italian Association for Cancer Research Special Program Molecular Clinical Oncology, 5 per mille #9965 and Investigator Grant #14511 to A.B.

## Author contribution

D.F., M.W., E.L., L.G., S.B., E.M., N.S., A.D.L., R.V., R.B., C.S., and C.D.L., performed experiments; M.T.S.B., performed the preparation of *Eμ-TCL1* mice; D.L., performed RNA-seq analysis; L.P., analyzed microarray, published datasets, and RNA-seq data; M. G.C. and L.M. designed and performed retinoic acid quantification experiments; M.P., L. B., B.G.F., E.C., A.C., D.L., and S.B.G. designed and performed immunohistochemistry and immunofluorescence staining on CLL tissue biopsies; F.C.C. and P.G. supervised the study and provided expertize with patient analysis; E.C. and L.S. provided CLL samples; D.F., M.W., E.L., and A.B. analyzed data, prepared figures, and wrote the manuscript; and A.B. directed the study. All authors discussed the results and provided comments for the manuscript.

## Additional information

**Competing interests:** The authors declare no competing interests.

