## [Peer Review File · Nature Communications]

Reviewers' comments:

Reviewer #1 (Remarks to the Author):

In this manuscript, Farinello et al. report on a retinoic acid-dependent crosstalk between stroma and chronic lymphocytic leukemia (CLL) cells promoting leukemia progression. By using a well established mouse model of CLL, they document that CLL cells induce the synthesis, and consequently cell signaling, of retinoic acid in stromal cells. Blockade of retinoic acid signaling inhibits several pro-survival functions of stromal cells, reinforcing the relevance of retinoic acid in CLL pathogenesis. The authors argue that targeting retinoic acid signaling may have therapeutic implications for CLL. The study is methodologically well conducted, the data are solid and novel, and the implications may be of relevance for a deeper understanding of stromal remodeling in CLL development. Few issues require further care, as listed below.

MAJOR ISSUES

1. Results, p. 6: By microarray analysis, the authors document that stromal cells activate several pathways of stromal remodeling, including the retinoic acid pathway, upon incubation with leukemic cells from the E μ -TCL1 mouse model. Were these results also confirmed upon incubation with primary CLL cells obtained from patients ex-vivo?
2. One main conclusion is that retinoic acid signaling strengthens stroma-leukemia interactions by promoting cellular adhesion. Is this conclusion based on one single stromal cell line model and one single CLL sample?
3. Overall, both in the text and in figures, it should be rendered clear when primary CLL cells are used versus E μ -TCL1 leukemic cells.
4. Consistent with results in mice models, the authors document the presence of CXCL13+ stromal cells also in the lymph nodes of CLL patients. Did the fraction of CXCL13+ cells vary with disease aggressiveness, refractoriness to therapy or histologic transformation?
5. The discussion should clearly highlight the limitations of this study and should acknowledge that the main body of the conclusions derive from results in mice models, that may not fully reflect CLL in humans.

MINOR ISSUES

1. The authors should be homogeneous in the acronyms adopted through the manuscript (eg: chronic lymphocytic leukemia is mostly, and correctly, addressed as CLL, but sometimes the acronym B-CLL occurs)
2. p. 7, l. 185: "ablation of the RA-signaling" should read as "ablation of RA-signaling"

Reviewer #2 (Remarks to the Author):

The manuscript by Farinello et al. describes the interactions between mesenchymal stroma and CLL and their potential biological and therapeutic implications.

The authors analyze the changes induced by the malignant lymphocytes in the mesenchymal stroma. They describe a number of these changes and focus their attention on changes that relate to retinoic acid (RA) biosynthesis and signaling in the stromal cells. These changes can be rescued by inhibition of retinoic acid signaling and can be reproduced by modulation of the retinoic acid pathway in stroma cells in the absence of CLL cells.

As it pertains to this part of their work, the experiments are well controlled and the data is very compelling. It would be of interest to further understand to what extent the induction of RA signaling in stroma cells depends on upregulation of RA biosynthesis machinery. Alternatively, free RA produced by CLL cells could directly induce upregulation of RA signaling in surrounding stroma cells in a paracrine manner that has been shown to take place in a variety of systems (take for instance gonads where mesonephros has high expression of ALDH1A1 and produces RA to induce meiosis in neighboring female germ cells) (Bowles J et al. Science, 2006). To this end, using either

genetically engineered stroma cells (ALDH1A1 knockout) or ALDH inhibitors (DEAB) may shed light into the contribution of autocrine vs. paracrine effects of RA in the mesenchymal stroma.

The authors set to test the biological implications of CLL-induced gene expression changes in the stroma cells on the CLL homeostasis. They describe a clever system approach in which CLL cells and stroma cells are forced to interact in a 3D culture and treated with BMS493 (a pan-RAR inhibitor). The way the data is presented here (mostly figure 2D), as well as the level of experimental detail provided in the methods is insufficient to draw any definitive conclusions. In this system, since both CLL cells and stroma cells are treated, it is likely that treatment with BMS493 affects both components of the system. More so, as expression of CXCR5 (the ligand for CXCL13) on CLL cells is likely to be decreased since it was shown to be directly controlled by RA signaling via RA-responsive elements in its promoter (Wang J and Yen A, MCB 2004). The experiment presented in figure 2C appears to indicate that the observed effect seen in these 3D cultures is at least in part due to changes in the stroma but even in this experimental set up, the stroma and CLL cells were treated together for about 16h prior to the readout. Since the protein kinetics of CXCR5 and CXCL13 is not clear, the observed effect could still be due to action of BMS493 on CLL cells and not on stroma. Nevertheless, it is very difficult to understand what the data in figure 2D and 2C represents. The way I read this data is that about 60% of CLL cells in these 3D cultures (80% in regular co-culture – figure 2C) treated with BMS493 were in the lymphoid aggregates. If so, what was the total number of CLL cells retrieved from these cultures (either as part of the aggregates or not part of the aggregates). If I misunderstood this data, please clarify what it represents.

Using in vivo models of either Eu-TCL1 CLL cells transplanted in WT mice or original Eu-TCL1 cells, the authors go on to show that similar induction of CXCL13 positive stroma cells take place during initiation and progression of the disease. The data is compelling that with increased tumor burden there is increased expression of CXCL13 in some population of stroma cells, though the exact population is not known. The clinical/biological implication is also not directly addressed in this manuscript but since CXCL13 is thought to be a chemoattractant for CLL cells, overexpression may be associated with increased homing of CLL cells into the lymphoid organs.

In the next set of experiments, presented mostly in figure 5, using three in vivo models the authors investigate to what extent modulation of RA signaling may play a role in either leukemogenesis or CLL homeostasis. Although the data is very clear, the results deserve careful interpretation especially given the role of retinoids in normal B-cell development.

The authors find that the engraftment of Eu-TCL1 CLL cells is significantly decreased, though it would be compelling to characterize the levels of CXCL13 ligand or CXCL13 positive stroma cells in these mice. The hypothesis being that VAD deficient mice would have relatively dysregulated CXCL13 expression/levels compared to control mice.

In the second in vivo model, the authors fed VAD diet to Eu-TCL mice prior to develop CLL and they notice that the VAD mice survive longer than control mice. In this model, I think it is particularly important to recognize the role of vitamin A in B cell differentiation and comment on the B cell development of these mice prior to disease as well as normal B cells in mice that already show signs of CLL.

For obvious reasons, the first two models have no therapeutic relevance but they do suggest that RA signaling plays an important role in disease homeostasis.

In a therapeutically more relevant model of disease, WT mice were engrafted with Eu-TCL1 cells and subsequently treated with BMS493. Analysis of various organs, including BM and PB show a relative decrease frequency of CLL cells. It is unclear if this decrease frequency has any impact on survival of these mice. Without this data, it would be speculative to imply that inhibition of RA signaling has therapeutic benefit in this model.

In conclusion, there is clear evidence that CLL cells change the surrounding stroma but the exact mechanism by which that happen remains unknown. It is also not clear if and how these changes in stroma have impact the malignant cells. Most of the in vitro data as well as all the in vivo data could be explained by direct effect of retinoids on the CLL clone and not their impact on stromal microenvironment. More so, recent data looking on cross-talk between malignant cells and

mesenchymal stroma suggest that malignant cells induce retinoid catabolism via upregulation of CYP26 (Alonso S et al. 2016). The data presented here appear to suggest the opposite with CLL cells upregulating RA biosynthesis in stroma. Can the author discuss potential explanations for these incongruences? If so, what is the net effect of CLL induced changes in stroma on RA homeostasis in these systems? Do mice with CLL for instance have higher levels of retinoic acid in their spleen compared to control?

In regard to using RAR inhibitors as therapeutic tools in CLL, this is in contrast with most data presented in human hematological malignancies. That being said, the data presented here comes to further confirm differences between mouse and human models and the role of retinoids in these models. To this end, data from human and mouse hematopoietic stem cells (HSCs) clearly shows that RA promotes expansion of mouse HSC (Cabezas-Wallscheid et al Cell 2017, Purton LE et al Blood 2000) and mouse AML-ETO leukemia stem cells while it differentiates human HSCs (Chute Jp et al 2006, Ghiaur G et al. 2013) and human AML.

In more general terms, using this one mouse model of CLL, though widely used in the literature may not reflect the heterogeneity of human disease. It would be of value of the authors could show that CLL cells from patients induce similar changes in stroma cells. Data like this would go a long way to confirm the validity of this mouse model in this particular biological finding.

Thank you for allowing me to participate in this review.

Reviewer #3 (Remarks to the Author):

This is an extremely well performed and described study of the role of retinoic acid in the development and progression of chronic lymphatic leukemia (CLL) in an animal model. Certain of the observed phenomena are also examined and appear to also relate to human CLL.

I have two questions that should be addressed in more detail.

1. What effects other than inhibition of RA stimulation does BMS293 have in vivo? Are there off-target effects that could influence the phenomena observed?
2. Which retinoic acid receptors are involved in the observed phenomena and in the induced effects?

We thank the Referees for their review and the helpful observations. Based on their constructive comments we have revised our manuscript by providing new data and clarifications, which we believe strengthen the main message, improved the quality of the manuscript, and address referees' major and minor concerns. To support the referees in the re-evaluation of this study, text changes and additions in the revised manuscript are shown in bold. Please find below a point-by-point response to each reviewer's comments.

Reviewer #1 (Remarks to the Author).

In this manuscript, Farinello et al. report on a retinoic acid-dependent crosstalk between stroma and chronic lymphocytic leukemia (CLL) cells promoting leukemia progression. By using a well established mouse model of CLL, they document that CLL cells induce the synthesis, and consequently cell signaling, of retinoic acid in stromal cells. Blockade of retinoic acid signaling inhibits several pro-survival functions of stromal cells, reinforcing the relevance of retinoic acid in CLL pathogenesis. The authors argue that targeting retinoic acid signaling may have therapeutic implications for CLL. The study is methodologically well conducted, the data are solid and novel, and the implications may be of relevance for a deeper understanding of stromal remodeling in CLL development. Few issues require further care, as listed below.

MAJOR ISSUES

1. Results, p. 6: *By microarray analysis, the authors document that stromal cells activate several pathways of stromal remodelling, including the retinoic acid pathway, upon incubation with leukemic cells from the *Eμ-TCL1* mouse model. Were these results also confirmed upon incubation with primary CLL cells obtained from patients ex-vivo?*

Reply: This is an important point that we have addressed by performing a new set of experiments using human primary CLL cells. We first performed the experiment by co-culturing 8 primary human CLL cases (7 stable and 1 progressive, the latter labelled Pt-08 in the histograms of Figure 1 below) with our murine stromal cell line, the same used for the microarray and RNA-seq analyses. The results indicate that, similar to what observed with murine *Eμ-TCL1* CLL cells, human CLL samples induce in stromal cells the expression, at different levels, of several genes related to retinoid-synthesis (*Rdh10*, *Aldh1a1*, *Cyp1b1*) and stroma/ECM remodelling (*αSMA*, *Prelp*, *Nidogen2*).

Figure 1. Murine stroma-specific gene signature induced by human CLL.

Human primary CLL cells were cultured on a layer of a murine stromal cell line (already used for generating microarray data) for 24 hrs, and the expression of stroma-specific genes was assessed by qPCR and expressed as relative mRNA levels.

We then compared our microarray data (of stromal cells cultured with murine CLL cells) to the gene expression profile recently published by Paggetti et al., Blood 2015.

In their work Paggetti et al, showed that CLL-derived exosomes induce gene signatures (e.g. α SMA also known as ACTA2) in a human stromal cell line, that are typical of cancer associated fibroblasts, a phenotype we have also found in our murine stromal cell line cultured with murine *E μ -TCL1* CLL cells (microarray data in Figure 1 and S1 of the manuscript). In addition, by comparing the two datasets, we noticed that both mouse and human CLL cells induce the expression (in stromal cells) of genes associated with inflammatory responses and interferon signalling (Figure 3 below, left panel). Moreover, similar to what we found (in Figure 1A manuscript) Paggetti et al. shows the down-regulation of signatures associated with cell cycle and mitosis (Figure 3 left panel below). Importantly, by comparing the two datasets, we also found that both mouse and human

CLL cells induce the up-regulation of similar (found in common) gene signatures; whereas only some of the down-regulated genes showed a similar trend in the two datasets.

Figure 3. Annotation of gene signatures induced by human CLL in human stromal cells

Annotations of stromal-gene signatures by human CLL cells revealed similarities between mouse and human (right). Fold change of genes commonly deregulated in stromal cells by mouse (our microarray data) and human CLL cells (from Paggetti et al. data sets).

Together these findings, strongly indicate that mouse CLL cells and human primary CLL cells induce similar changes in stromal cells, indicating functional similarities between mouse and human CLL cells.

2. One main conclusion is that retinoic acid signaling strengthens stroma-leukemia interactions by promoting cellular adhesion. Is this conclusion based on one single stromal cell line model and one single CLL sample?

Reply: We apologize for not being clear on this. Our microarray data were obtained using mRNA from 4 independent experiments performed using primary murine *Eμ-TCL1* CLL preparations established from 4 distinct *Eμ-TCL1* transgenic mice with similar % of leukemia in the spleen, and cultured on our murine stromal cell line (mSSC). We recently published a manuscript (Lenti et al. JCI 2016) in which we describe another spleen stromal cell line that responds to retinoids and to retinoid-signaling inhibition (BMS493) in the same manner as the cell line used in our manuscript, indicating that the response of different murine stromal cells to retinoids appear to be conserved.

3. Overall, both in the text and in figures, it should be rendered clear when primary CLL cells are used versus *Eμ-TCL1* leukemic cells.

Reply: We apologize for the lack of clarity on this. We have revised the manuscript accordingly and have indicated more clearly when human or murine CLL were used.

4. Consistent with results in mouse models, the authors document the presence of CXCL13+ stromal cells also in the lymph nodes of CLL patients. Did the fraction of

CXCL13+ cells vary with disease aggressiveness, refractoriness to therapy or histologic transformation?

Reply: We have now extended our analysis to 2 stables, 4 progressive and 2 transformed CLL cases (Figure 4 below). These results indicate that CXCL13 signal increases during disease transformation, although this should be studied in a larger cohort of cases to further consolidate the findings. **We have now added these new findings in Suppl. Figure 3.**

Figure 4. CXCL13 in human lymph node CLL biopsies.

Bright field images of human CLL lymph node sections immunostained for CXCL13 (brown). Graph represents the average number of CXCL13+ cells from five high power fields (HPF) for each sample. Original magnification: 200x.

5. The discussion should clearly highlight the limitations of this study and should acknowledge that the main body of the conclusions derive from results in mice models, that may not fully reflect CLL in humans.

Reply: We have now highlighted in the main text that most of the data were obtained from mouse models.

MINOR

ISSUES

1. The authors should be homogeneous in the acronyms adopted through the manuscript (eg: chronic lymphocytic leukemia is mostly, and correctly, addressed as CLL, but sometimes the acronym B-CLL occurs)

Reply: We have revised the text and adopted the same acronyms through the manuscript

2. p. 7, l. 185: “ablation of the RA-signaling” should read as “ablation of RA-signaling”

Reply: We have revised the text accordingly.

Reviewer #2 (Remarks to the Author):

The manuscript by Farinello et al. describes the interactions between mesenchymal stroma and CLL and their potential biological and therapeutic implications. The authors analyze the changes induced by the malignant lymphocytes in the

mesenchymal stroma. They describe a number of these changes and focus their attention on changes that relate to retinoic acid (RA) biosynthesis and signaling in the stromal cells. These changes can be rescued by inhibition of retinoic acid signaling and can be reproduced by modulation of the retinoic acid pathway in stroma cells in the absence of CLL cells.

1. As it pertains to this part of their work, the experiments are well controlled and the data is very compelling. It would be of interest to further understand to what extent the induction of RA signaling in stroma cells depends on upregulation of RA biosynthesis machinery. Alternatively, free RA produced by CLL cells could directly induce upregulation of RA signaling in surrounding stroma cells in a paracrine matter that has been shown to take place in a variety of systems (take for instance gonads where mesonephros has high expression of ALDH1A1 and produces RA to induce meiosis in neighbouring female germ cells) (Bowles J et al. Science, 2006). To this end, using either genetically engineered stroma cells (ALDH1A1 knockout) or ALDH inhibitors (DEAB) may shed light into the contribution of autocrine vs. paracrine effects of RA in the mesenchymal stroma.

Reply: This is an important point, and indeed our microarray data and validation qPCR experiments (Figure 1 and S1 of the manuscript) indicate that key genes encoding enzymes involved in the synthesis of retinoic acid (*Aldh1a1*, *Aldh3b1* and *Cyp1b1*) are upregulated upon co-culture with murine CLL preparations established from different *E μ -TCL1* mice, thus indicating that stromal cells are capable to produce retinoids. In addition, we have also performed the experiment suggested by the reviewer using the ALDH inhibitor DEAB. We analysed 5 new murine *E μ -TCL1* CLL samples and 2 control

MOUSE	STRAIN	ALDEFLUOR+/CD19+	ALDEFLUOR+/CD19+ with DEAB	ALDEFLUOR+/CD19+CD5+	ALDEFLUOR+/CD19+CD5+ with DEAB
WT1	C57BL6	0.999	0.313	3.17	1.2
WT2	C57BL6	0.135	0.0215	0.532	0.233
#439	Eμ-TCL1	5.45	0.011	20.9	0.094
#419	Eμ-TCL1	14	1.21	30.3	0.52
#411	Eμ-TCL1	4.54	0.0807	10.2	0.0861
#468	Eμ-TCL1	8.93	0.0219	19.2	0.104
#381	Eμ-TCL1	15.6	1.2	17.5	0.3

Figure 5. A fraction of leukemic cells possesses ALDH activity and can produce retinoids. *E μ -TCL1* leukemic cells isolated from the spleen of 5 different leukemia-bearing mice and two control mice were assessed for Aldefluor activity in the presence or absence of the ALDH control inhibitor (DEAB). One representative analysis is shown. Note that in wild-type (WT) Aldefluor+CD19+ B cells are below 1%.

mice, and found that a fraction of CD19⁺CD5⁺ E μ -TCL1 leukemic cells - but not control CD19⁺ B cells - are Aldefluor⁺ (Figure 5 above), indicating that they possess the machinery to convert retinaldehyde into retinoid acid, and thus could secrete retinoids as hypothesized (These data have been included in new Figure S2 of the manuscript).

In line with this, we also provide to the reviewer new data indicating that both the murine stromal cell line and two human primary stromal cells derived from CLL patients (one from spleen and one from lymph nodes) are capable to activate RA signaling in responding RARE reporter cells (Figure 6 below). These findings together with our previous data showing that CLL cells are capable to stimulate RA-signaling in the same system (Figure 1 panel B of the manuscript) strongly suggest that both cell types can secrete/produce retinoids.

We have now discussed this point based on the new data. We also modified the discussion to integrate these data.

Figure 6. Stromal cells are capable to induce RA signalling in responder cells.

Murine or human stromal cells were cultured for 48 hrs on top of a retinoic acid reporter cell line (RARE-LacZ) expressing β -galactosidase under a Retinoic Acid Responsive Element (RARE). β -gal activity, indicative of RA-signaling activation was assessed as described in material and methods.

2. The authors set to test the biological implications of CLL-induced gene expression changes in the stroma cells on the CLL homeostasis. They describe a clever system approach in which CLL cells and stroma cells are forced to interact in a 3D culture and treated with BMS493 (a pan-RAR inhibitor). The way the data is presented here (mostly figure 2D), as well as the level of experimental detail provided in the methods is insufficient to draw any definitive conclusions. In this system, since both CLL cells and stroma cells are treated, it is likely that treatment with BMS493 affects both components of the system. More so, as expression of CXCR5 (the ligand for CXCL13) on CLL cells is likely to be decreased since it was shown to be directly controlled by RA signaling via RA-responsive elements in its promoter (Wang J and Yen A, MCB 2004).

The experiment presented in figure 2C appears to indicate that the observed effect seen in these 3D cultures is at least in part due to changes in the stroma but even in this experimental set up, the stroma and CLL cells were treated together for about 16h prior to the readout.

Since the protein kinetics of CXCR5 and CXCL13 is not clear, the observed effect could still be due to action of BMS493 on CLL cells and not on stroma. Nevertheless, it is very difficult to understand what the data in figure 2D and 2C represents. The way I read this data is that about 60% of CLL cells in these 3D cultures (80% in regular co-culture – figure 2C) treated with BMS493 were in the lymphoid aggregates. If so, what was the total

number of CLL cells retrieved from these cultures (either as part of the aggregates or not part of the aggregates). If I misunderstood this data, please clarify what it represents.

Reply: We apologize for the lack of clarity in the scheme of Figure 2; we have now revised the scheme in new Figure 2 and the text of the manuscript to explain better the way the experiments in panel 2C-D were done. Basically, all the experiments described in Figure 2C were performed using conventional bi-dimensional cultures. In these experiments, only stromal cells were treated with BMS493 for 72 hrs, then the media was replaced, and *E μ -TCL1* leukemic cells were added to the culture for the remaining 16 hrs, in the absence of the inhibitor. Thus, during the co-culture (Figure 2C on the manuscript), CLL cells were not exposed to BMS493. Cell adhesion of CLL cells to stroma was evaluated by counting the number of CLL cells attached to the monolayer of stroma and the result expressed in % of adherent cells relative to control. We think that under these conditions the observed effect (reduced adhesion of leukemia to stroma) is likely to be due to an intrinsic alteration of stromal cells upon RA-signaling inhibition. In support of this, our RNA-seq data (Figure 2A of the manuscript) of only stromal cells treated with BMS493 indicate that RA-signaling inhibition reduces expression of adhesion molecules, including genes involved in cell-ECM-receptors interactions.

After having demonstrated that treatment of stromal cells with the RA-signaling inhibitor reduces adhesion of CLL cells in conventional bi-dimensional cultures, we developed a 3D leukemic microenvironment suitable to recapitulate interactions that may occur in vivo in lymphoid tissues. This approach was chosen with the purpose of recapitulating the effect of the treatment in vivo, where the inhibitor would target both stroma and CLL cells. In this model, treatment with BMS493 causes a 40% reduction of leukemia aggregation (Figure 2D of the manuscript). This was calculated by counting the total number of cells retrieve from one organoid using MACSQuant Flow Cytometer. The average number of cells obtained from each BMS493-treated organoid was 45965, whereas in control organoids the number of cells was 73715.

We think that these two complementary approaches strongly indicate that inhibition of RA-signaling reduces cell-cell adhesion and interactions, and that inhibition of this signalling pathway in stromal cells largely contribute to the observed phenomena. Further supporting this notion is the fact that BMS493 treatment in mice significantly reduces the accumulation of leukemia in secondary lymphoid tissues (Figure 5 panel C and D of the manuscript).

*2. Using in vivo models of either *E μ -TCL1* CLL cells transplanted in WT mice or original *E μ -TCL1* cells, the authors go on to show that similar induction of CXCL13 positive stroma cells take place during initiation and progression of the disease. The data is compelling that with increased tumor burden there is increased expression of CXCL13 in some population of stroma cells, though the exact population is not known. The clinical/biological implication is also not directly addressed in this manuscript but since CXCL13 is thought to be a chemoattractant for CLL cells, overexpression may be associated with increased homing of CLL cells into the lymphoid organs.*

Reply: We apologize for not being clear on this. Our data in Figures 4C indicate that CXCL13+ cells are follicular stromal cells and that a fraction of these cells are indeed CD35+ follicular dendritic cells (FDCs) that co-express CXCL13 (Figure S3A). As the reviewer correctly pointed out, we have not addressed the clinical/biological implications of targeting CXCL13 since previous work by Heinig et al. (Ref 25) already showed that treatment with an anti-CXCL13 antibody reduces the splenic accumulation of leukemic cells in the *E μ -TCL1* model, thus corroborating the reviewer's hypothesis that CXCL13

overexpression may be associated with increased homing of CLL cells into the lymphoid organs.

3. *In the next set of experiments, presented mostly in figure 5, using three in vivo models the authors investigate to what extent modulation of RA signaling may play a role in either leukemogenesis or CLL homeostasis. Although the data is very clear, the results deserve careful interpretation especially given the role of retinoids in normal B-cell development. The authors find that the engraftment of E μ -TCL1 CLL cells is significantly decreased, though it would be compelling to characterize the levels of CXCL13 ligand or CXCL13 positive stroma cells in these mice. The hypothesis being that VAD deficient mice would have relatively dysregulated CXCL13 expression/levels compared to control mice.*

Reply: The hypothesis of the reviewer is indeed correct, and previous already published work published by Suzuki et al., *Immunity* 2010 demonstrated that VAD mice have reduced splenic CXCL13 levels as compared to controls. In addition, it has also been demonstrated that loss of CXCL13 causes defect in B-cell homing to lymphoid tissues (Gunn et al., *Nature* 1998; and Ansel et al., *Immunity* 2002). Thus, it is likely that reduce CXCL13 levels due to the loss of appropriate RA signalling may, at least in part, affect homing of normal and malignant E μ -TCL1 CLL cells.

In addition, we have also analysed CXCL13 levels in the spleen of mice engrafted with E μ -TCL1 CLL cells and treated or not with BMS493. The results (Figure 7 below) showed that CXCL13 signal is less prominent in leukemic mice treated with BMS493. However, this phenotype could also be due the reduced number of leukemic cells infiltrating the spleen and that could stimulate CXCL13 in stromal cells via the LT β R engagement.

Figure 7. Reduced CXCL13 domains in mice treated with the RA-signaling inhibitor BMS493. Representative mosaic confocal images of the spleen from mice transplanted with E μ -TCL1 leukemic cells and treated with BMS493 or DMSO (vehicle) and stained for CXCL13.

4. *In the second in vivo model, the authors fed VAD diet to E μ -TCL1 mice prior to develop CLL and they notice that the VAD mice survive longer than control mice. In this model, I think it is particularly important to recognize the role of vitamin A in B cell differentiation and comment on the B cell development of these mice prior to disease as well as normal B cells in mice that already show signs of CLL. For obvious reasons, the first two models have no therapeutic relevance but they do suggest that RA signalling plays an important*

role in disease homeostasis. In a therapeutically more relevant model of disease, WT mice were engrafted with $E\mu$ -TCL1 cells and subsequently treated with BMS493. Analysis of various organs, including BM and PB show a relative decrease frequency of CLL cells. It is unclear if this decrease frequency has any impact on survival of these mice. Without this data, it would be speculative to imply that inhibition of RA signaling has therapeutic benefit in this model.

Reply: We agree with the reviewer and we have acknowledged in the discussion the possible role of retinoids in B-cell development. We believe that this notion further strengthens the idea that not only retinoids are involved in the biology of B-cells, but also that inhibition of retinoid signalling could be exploited as a strategy to target B-cell malignancies.

Based on the reviewer comments, we have now performed a survival curve upon BMS493 treatment. The results show that wild-type mice engrafted with $E\mu$ -TCL1 cells, and five days later treated with BMS493 (n=10) survive significantly longer as compared to controls (DMSO) (n=10) (Figure 8 below). Consistent with this, echographical measurement of the spleen size during disease progression (day 35) revealed that treatment with BMS493 significantly reduced infiltration of leukaemia cells in the spleen. This new finding is in agreement with our original data in Figure 5C showing that BMS493 treatment reduces leukaemia infiltration not only in the spleen but also in PB and BM. **These new data have been included in Figure 5 of the manuscript.**

Figure 8. The RA-signaling inhibitor BMS493 prolongs survival.

(A) Survival curve of mice treated with the RA-signaling inhibitor BMS493 or control vehicle (top). (B) Echographical measurement of the spleen size performed at day 35 of disease progression (bottom).

5. In conclusion, there is clear evidence that CLL cells change the surrounding stroma but the exact mechanism by which that happen remains unknown. It is also not clear if and how these changes in stroma have impact the malignant cells. Most of the *in vitro* data as well as all the *in vivo* data could be explained by direct effect of retinoids on the CLL clone and not their impact on stromal microenvironment.

Reply: We thank the reviewer for this reasoning but we believe that our data do suggest that RA-signaling inhibition does alter the stromal microenvironment per se. Indeed, our RNA-seq data (Figure 5 of the manuscript) clearly demonstrate that inhibition of RA-signalling in stromal cells causes deregulation of gene signatures belonging to different

pathways and these include adhesion, ECM-CLL interactions, chemokines and migration, and we validated these findings in adhesion assay as well (Figure 2 of the manuscript). Thus, retinoid-signalling inhibition does impact on the biology of stromal cells. This notion is further supported by the fact that antagonizing RA-signaling in stromal cells restores the expression a large fraction of genes (also involved in adhesion) that were induced by leukemic cells (Figure 2B of the manuscript).

[REDACTED]

interactions, and thus affects cell viability by reducing the protective effect of the stroma.

[REDACTED]

[REDACTED]

6. *More so, recent data looking on cross-talk between malignant cells and mesenchymal stroma suggest that malignant cells induce retinoid catabolism via upregulation of CYP26 (Alonso S et al. 2016). The data presented here appear to suggest the opposite with CLL cells upregulating RA biosynthesis in stroma. Can the author discuss potential explanations for these incongruences? If so, what is the net effect of CLL induced changes in stroma on RA homeostasis in these systems? Do mice with CLL for instance have higher levels of retinoic acid in their spleen compared to control?*

Reply: In the paper mentioned by the reviewer, Alonso et al., describes a model in which stromal cells are induced by Multiple Myeloma (MM) to express CYP16B1, an enzyme involved in RA degradation. The mechanism responsible for this effect remains unclear, although it is likely that retinoids either secreted by MM cells or by cells of the microenvironment induce CYP26B1. These enzymes are indeed under the control of RA-

nuclear receptors and are known to be expressed upon RA exposure (Zolfaghari et al., JCB 2014; Takeuchi et al., 2011).

Indeed, our data in Figure 2A of the manuscript show that inhibition of RA-signaling causes a significant reduction of *Cyp26b1* expression, indicating that CYP26 enzymes are under the control of retinoids. In the case of Alonso paper, it is likely that an RA-low microenvironment sustained by CYP26B1 promotes the activation of pathways associated with drug resistance. In the case of MM, CYP26b1 prevents retinoids to act as pro-differentiating agents, a mechanism different from what we propose and that we have not investigated in our work.

7. In regard to using RAR inhibitors as therapeutic tools in CLL, this is in contrast with most data presented in human hematological malignancies. That being said, the data presented here comes to further confirm differences between mouse and human models and the role of retinoids in these models. To this end, data from human and mouse hematopoietic stem cells (HSCs) clearly shows that RA promotes expansion of mouse HSC (Cabezas-Wallscheid et al Cell 2017, Purton LE et al Blood 2000) and mouse AML-ETO leukemia stem cells while it differentiates human HSCs (Chute Jp et al 2006, Ghiaur G et al. 2013) and human AML.

Reply: We agree with the reviewer that retinoids have multiple effects, and that these may differ between mouse and human. However, it is important to note that the use of retinoids in clinical setting is successful exclusively for the differentiation of AML leukemias. In this context, retinoids are used as pro-differentiating agents through a mechanism that involves the degradation of the PML-RAR α oncoprotein, and the unleashing of differentiation programs. In the case of CLL there is no evidence that retinoids induce differentiation programs similar to AML.

In light of our findings, we propose that inhibition of RA-signaling may represent a novel strategy to target stroma-leukemic interactions through multiple pathways (different from cellular differentiation) involving adhesion, migration, and ECM-remodeling (Figure 2A).

8. In more general terms, using this one mouse model of CLL, though widely used in the literature may not reflect the heterogeneity of human disease. It would be of value if the authors could show that CLL cells from patients induce similar changes in stroma cells stroma-hCLL culture exp. Data like this would go a long way to confirm the validity of this mouse model in this biological finding.

Reply: We agree that the mouse model may not reflect the heterogeneity of human CLL and we have further highlighted this in the discussion (page XX). In addition, we are now presenting additional data related to human CLL to show the relevance of our findings and the similarities between mouse and human CLL also in response to a similar request from reviewer 1 (please see first reply on page 3). Thus, we believe that the mouse and human data share several important affinities, although future work will be necessary to establish more precisely the role of retinoid metabolism in human CLL.

Reviewer #3 (Remarks to the Author).

This is an extremely well performed and described study of the role of retinoic acid in the development and progression of chronic lymphatic leukemia (CLL) in an animal model. Certain of the observed phenomena are also examined and appear to also relate to human CLL.

I have two questions that should be addressed in more detail.

1. What effects other than inhibition of RA stimulation does BMS293 have in vivo? Are there off-target effects that could influence the phenomena observed?

Reply: This is an important point. Though we cannot completely exclude the possibility of off-target effects, the relevant phenomenon observed in our experiments seem to be on-target. Accordingly, as discussed above, upon BMS493 treatment we have observed reduced CXCL13 distribution and this in line with *Cxcl13* being a target of RA signaling as previously reported in vitro and in vivo (van de Pavert, *Nat Immunology* 2009). van de Pavert et al., showed that CXCL13 promoter contains a RARE element, thus suggesting that this gene is likely a direct target. Suzuki et al., *Immunity* 2010 also showed that vitamin A deficient mice have low CXCL13 in the spleen, thus indicating that the reduction in CXCL13 upon BMS493 is not an off-target effect. In addition, in a different work, van de Pavert et al., *Nature* 2014, showed that treatment with BMS493 leads to a reduction lymph node dimension. This phenotype, was also recapitulated by genetic ablation of RA-signaling in vivo. Given this result and considering that we used BMS493 at the same concentration used in van de Pavert et al., *Nature* 2014 (5mg/kg), it is unlikely that the phenomenon observed was caused by off-target effects. Off-target effects have been documented in chicken studies in which beads were soaked with much higher concentrations of RA agonists/antagonists.

2. Which retinoic acid receptors are involved in the observed phenomena and in the induced effects?

Reply: We have addressed this point by analysing the expression of different RA nuclear receptors in the splenic stromal cell line used for the microarray and RNA-seq data (Figure 10 below). The results showed that several RA nuclear receptor isoforms are expressed, but that BMS493 treatment significantly deregulates *Rarβ*, and to some extent *Rarγ* isoforms. Although these receptors are clearly modulated by the BMS493 treatment, and thus may be involved in the underlying phenomena, it is also possible that other RA nuclear receptors are involved which may not be transcriptionally controlled by retinoids but still could play a role in modulating downstream pathways.

Figure 10. Expression of Retinoic Acid nuclear receptors in mouse stromal cell line. Expression of RA nuclear receptor isoforms in stromal cells treated with or without the RA-signaling inhibitor BMS493 for 48 hrs.

REVIEWERS' COMMENTS:

Reviewer #1 (Remarks to the Author):

The authors have adequately addressed all the issues that had been raised, and I have no further comments. The additional data with primary CLL cells are very convincing and significantly add to the manuscript in the perspective of translational medicine.

Reviewer #2 (Remarks to the Author):

The authors addressed all the issues raised by this reviewer and even though, we may disagree on the interpretation of some of their findings, I believe that the additional data presented together with the discussions around the interpretation of the data, make this revised version of the manuscript highly pertinent to the field and of extremely high quality.

Reviewer #3 (Remarks to the Author):

The authors have adequately addressed my two initial questions.